# Cells recognize osmotic stress through liquid–liquid phase separation lubricated with poly(ADP-ribose)

Kengo Watanabe ⬡ [1✉], Kazuhiro Morishita[1], Xiangyu Zhou[1], Shigeru Shiizaki[1], Yasuo Uchiyama[2], Masato Koike ⬡ [3], Isao Naguro ⬡ [1] & Hidenori Ichijo ⬡ [1✉]

Cells are under threat of osmotic perturbation; cell volume maintenance is critical in cerebral edema, inflammation and aging, in which prominent changes in intracellular or extracellular osmolality emerge. After osmotic stress-enforced cell swelling or shrinkage, the cells regulate intracellular osmolality to recover their volume. However, the mechanisms recognizing osmotic stress remain obscured. We previously clarified that apoptosis signal-regulating kinase 3 (ASK3) bidirectionally responds to osmotic stress and regulates cell volume recovery. Here, we show that macromolecular crowding induces liquid-demixing condensates of ASK3 under hyperosmotic stress, which transduce osmosensing signal into ASK3 inactivation. A genome-wide small interfering RNA (siRNA) screen identifies an ASK3 inactivation regulator, nicotinamide phosphoribosyltransferase (NAMPT), related to poly(ADP-ribose) signaling. Furthermore, we clarify that poly(ADP-ribose) keeps ASK3 condensates in the liquid phase and enables ASK3 to become inactivated under hyperosmotic stress. Our findings demonstrate that cells rationally incorporate physicochemical phase separation into their osmosensing systems.

[1] Laboratory of Cell Signaling, Graduate School of Pharmaceutical Sciences, The University of Tokyo, Tokyo 113-0033, Japan. [2] Department of Cellular and Molecular Neuropathology, Juntendo University Graduate School of Medicine, Tokyo 113-8421, Japan. [3] Department of Cell Biology and Neuroscience, Juntendo University Graduate School of Medicine, Tokyo 113-8421, Japan. ✉email: kwatanabe@15.alumni.u-tokyo.ac.jp; ichijo@mol.f.u-tokyo.ac.jp

When a difference between intracellular and extracellular osmolality develops, cells inevitably become swollen or shrunken following osmotically driven water flow. The abnormal cellular osmoregulation leads to deteriorated pathophysiological conditions observed in cerebral edema, inflammation, cataracts and aging[1–6]. Basically, homeostasis in cell volume is vital for cellular activities and cells have a defense system against the disastrous osmotic stress; cells immediately excrete or intake ions and small organic solutes after hypoosmotic or hyperosmotic stresses, respectively, and recover their volume within minutes to hours by controlling ion channels and transporters[7–9]. Many electrophysiological and pharmacological studies have contributed to the accumulation of knowledge about the effector molecules in cell volume regulation. In contrast, the mechanisms sensing osmotic stress to induce cell volume recovery remain unclear, especially in mammalian cells. Similar to the osmosensors proposed in bacteria, yeasts and plants, mechanical changes in/on the cell membrane have recently drawn attention in mammalian cells; for example, membrane stretching under hypoosmotic stress activates mechanosensitive channels, such as the transient receptor potential channel V4[10,11]. This mechanism can be illustrated by an easy-to-understand signaling schematic with arrows directed from the extracellular side to the intracellular side. However, osmotic stress perturbs not only the cell membrane but also the intracellular ion strength/concentration and macromolecular crowding[7,8]; therefore, the existence of intracellular osmosensors may currently be underestimated.

In intracellular space, numerous biomolecules are crowded but marvelously organized for cellular activities. As a fundamental strategy, cells leverage compartmentation to regulate specific biomolecules spatiotemporally. Among the intracellular compartments, biomolecular condensates are quite unique. Unlike the classic organelles surrounded with lipid bilayers, biomolecular condensates are membraneless but concentrate appropriate biomolecules with phase boundary; in other words, biomolecular condensates are phase-separated from their outside space under the laws of thermodynamics[12–15]. Biomolecular condensates take various material properties, such as liquid, gel and solid, dependently on constituent biomolecules and environmental conditions including temperature, pH, ion strength/concentration and macromolecular crowding. Besides, biomolecular condensates are not static compartments; when the concentration or state of biomolecules or the environmental condition alters deeply enough to surpass the threshold, biomolecular condensates will undergo phase transition (e.g., from liquid to solid) to reach energetically desirable states. Note that, in perspective of whole cellular system, the phase transition can also induce the appearance and disappearance of biomolecular condensates. Therefore, biomolecular condensates ideally provide the dynamic and rapid spatiotemporal control of biomolecules. Studies on biomolecular condensates have been rapidly blooming just for the past decade. Nevertheless, lots of questions remain to be elucidated[12–15]; for instance, physicochemical theories and in vitro experiments suggest that macromolecular crowding is a driving force for the phase separation of biomolecular condensates, but the cellular system is detached from the classic thermodynamic equilibrium and the cell-based investigations have begun just recently[16]. Like the scientific history that every development of microscopy drastically advanced biology, the "lens" of phase separation would bridge spatiotemporal gaps of understanding between cell biology and molecular biology.

We previously reported that apoptosis signal-regulating kinase 3 (ASK3; also known as MAP3K15) is phosphorylated and activated under hypoosmotic stress and conversely dephosphorylated and inactivated under hyperosmotic stress[17]. In addition to the rapid, sensitive and reversible nature, this bidirectional response of ASK3 orchestrates proper cell volume recovery under both hypoosmotic and hyperosmotic stresses[18]. Therefore, we conceived the idea that the elucidation of ASK3 regulation under osmotic stress would lead to the clarification of a general mammalian osmosensing system.

In this study, we report that ASK3 forms liquid droplets under hyperosmotic stress, which is necessary for ASK3 inactivation. Moreover, by utilizing a genome-wide small interfering RNA (siRNA) screen, we reveal that poly(ADP-ribose) (PAR) maintains the liquidity of ASK3 droplets for ASK3 inactivation. Our findings demonstrate that cells recognize osmotic stress through liquid–liquid phase separation (LLPS) of ASK3 with the support of PAR.

## Results

**Hyperosmotic stress induces ASK3 condensation through liquid–liquid phase separation.** Through analyses of ASK3, we found that the subcellular localization of ASK3 drastically changes under hyperosmotic stress: a part of ASK3 diffuses throughout the cytosol, while the other forms granule-like structures, ASK3 condensates (Fig. 1a). The number of ASK3 condensates increased in a hyperosmolality strength-dependent manner (Fig. 1b) and gradually decreased several dozen minutes after hyperosmotic stress (Supplementary Fig. 1a, b), which corresponds to the time range of cell volume recovery[9,18]. In addition to mannitol-supplemented medium, sodium chloride-supplemented medium induced a similar pattern of ASK3 localization (Supplementary Fig. 1c, d), suggesting that hyperosmolality causes ASK3 condensates. Counterintuitively, the size of condensates was inversely associated with hyperosmolality (Fig. 1b). In fact, a simple computational model for protein diffusion and clustering in a two-dimensional grid space[19] predicted that the size of ASK3 clusters would increase when the grid space was reduced to mimic cell shrinkage under hyperosmotic stress (Supplementary Fig. 2a, b; see also Supplementary Discussion). However, there are abundant macromolecules in cells[20]; and we modified the model by adding obstacles to include effects of macromolecular crowding (Fig. 1c and Supplementary Note). Our simulation results demonstrated that decreasing the grid space progressively increases both the number and size of ASK3 clusters, while further decreasing the grid space eventually reduces the size of clusters (Fig. 1d, e, Supplementary Movie 1 and Supplementary Discussion), implying that the existence of macromolecular crowding is critical for ASK3 condensates under hyperosmotic stress in cells.

Further characterization revealed that ASK3 condensates are colocalized with a marker of neither early endosomes nor lysosomes (Supplementary Fig. 1e, f). Transmission electron microscopy (TEM) analysis with the immunogold-labeling technique revealed that ASK3 condensates are membraneless structures (Fig. 2a). Although stress granules (SGs) and P-bodies are known membraneless structures under extreme hyperosmotic conditions[21,22], markers of neither structure were found to be colocalized with ASK3 condensates (Supplementary Fig. 1g). Upon observing in 1-s intervals, we found that ASK3 condensates appear just seconds after hyperosmotic stress, which is much faster than in the cases of SGs and P-bodies, and that ASK3 condensates dynamically move around and fuse with each other (Fig. 2b, Supplementary Movie 2). Furthermore, our computational model predicted that the shrinkage-induced clusters gradually disappear when the grid space is reverted back to the large grid space (Fig. 2c, d, Supplementary Movie 3 and Supplementary Discussion), and we observed the similar kinetics of reversibility in cells (Fig. 2e, f). Interestingly, our model also predicted a transient increase in the size of clusters just after the

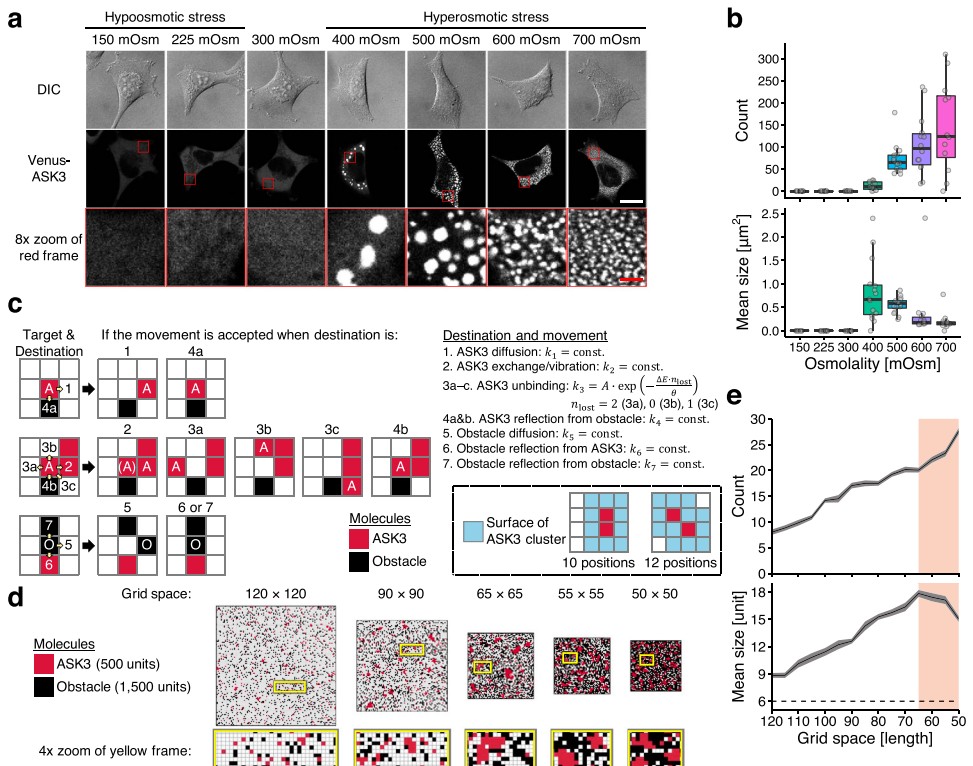

**Fig. 1 ASK3 forms macromolecular crowding-driven condensates under hyperosmotic stress. a, b** Subcellular localization of ASK3 5 min after osmotic stress in Venus-ASK3-stably expressing HEK293A (Venus-ASK3-HEK293A) cells. Hypoosmotic stress: ultrapure water-diluted medium, hyperosmotic stress: mannitol-supplemented medium, DIC: differential interference contrast, white bar: 20 μm, red bar: 2.5 μm. Data: center line = median; box limits = $[Q_1, Q_3]$; whiskers = [max(minimum value, $Q_1 - 1.5 \times IQR$), min(maximum value, $Q_3 + 1.5 \times IQR$)], where $Q_1$, $Q_3$ and IQR are the first quartile, the third quartile and the interquartile range, respectively; n = 12 (700 mOsm), 14 (600 mOsm), 15 (400 mOsm), 16 (150 mOsm, 225 mOsm, 300 mOsm and 500 mOsm) cells pooled from four independent experiments. Note that the signal intensity of DIC cannot be compared among the images. **c** Schematic diagram of a computational model for protein diffusion and clustering in a two-dimensional grid space. Red squares: ASK3 units, black squares: obstacles, pale yellow arrows: potential movements, blue squares: surface positions of the clusters. See Supplementary Note for the full description of the model. **d, e** A computational simulation for the relationship between the grid space and the number/size of ASK3 clusters. Results after $5 \times 10^6$ steps in the rejection kinetic Monte Carlo (rKMC) method at each grid space are presented. Red shading: the assumed range corresponding to hyperosmotic stress (details in Supplementary Discussion), dashed line: the minimum of ASK3 clusters definition. Data: mean ± SEM, n = 18 simulations.

grid space expansion, which was in good agreement with the cell-based experiments. To further address the dynamics of ASK3 condensates, we established a fluorescence recovery after photobleaching (FRAP) assay for the rapidly moving condensates and found that ASK3 condensates display not complete but significant FRAP (Fig. 2g, h), indicating that ASK3 molecules in the condensates are interchanged with those in the cytosol.

Given these characteristics, we concluded that ASK3 condensates are liquid-demixing droplets induced by LLPS[12–15]. Indeed, according to soft matter physics, there are two modes of LLPS, "nucleation and growth" and "spinodal decomposition", and we observed the spinodal decomposition-like pattern of ASK3 as a rare case (Supplementary Fig. 1h). In addition, crowding reagents, such as Ficoll and polyethylene glycol (PEG), induced ASK3 condensates in vitro (Fig. 2i, j). Although the intracellular ion strength and concentration are altered under hyperosmotic stress in cells, the change in sodium chloride concentration did not induce ASK3 condensates in vitro, suggesting again that macromolecular crowding but not ion strength is a critical driving force for the formation of ASK3 condensates under hyperosmotic stress. Of note, the condensates produced by our in vitro assays are solid-like because we could not observe their FRAP, but the spherical shape of condensate implies that the condensates were formed by LLPS and rapidly matured from liquid state into solid-like state[15,23].

**C-terminus coiled-coil domain and low-complexity region are required for ASK3 condensation followed by ASK3 inactivation under hyperosmotic stress.** To clarify the significance of ASK3 condensation in cells, we first generated ASK3 mutants that are unable to condense. While ASK3 ΔN mutant normally condensed, ASK3 ΔC mutant lost the ability to condense under hyperosmotic stress (Fig. 3a, b). Moreover, ASK3 C-terminus (CT) fragment formed condensates even under isoosmotic conditions, while ASK3 N-terminus and kinase domain (KD) fragments did not exhibit the ability to condense, suggesting that the CT region of ASK3 is necessary and sufficient for ASK3 condensation. In the ASK3 CT region, five distinctive regions are bioinformatically predicted (Supplementary Fig. 3a): two intrinsically disordered regions (IDRs; Supplementary Fig. 3b), one coiled-coil (CC) domain, one sterile alpha motif (SAM) domain[24] and one low-complexity region (LCR). Between ASK3 CT fragment mutants with these deletions, the CTΔCC, CTΔSAM and CTΔLCR mutants exhibited reduced condensation ability compared with the original CT fragment (Supplementary Fig. 3c). Considering the location of LCR within SAM domain, we next deleted both CC domain and SAM domain or LCR from the original ASK3 CT fragment and found that the CTΔCCΔSAM and CTΔCCΔLCR mutants are unable to condense even under hyperosmotic stress (Supplementary Fig. 3d), suggesting that both CC domain and LCR contribute to ASK3 condensation. In

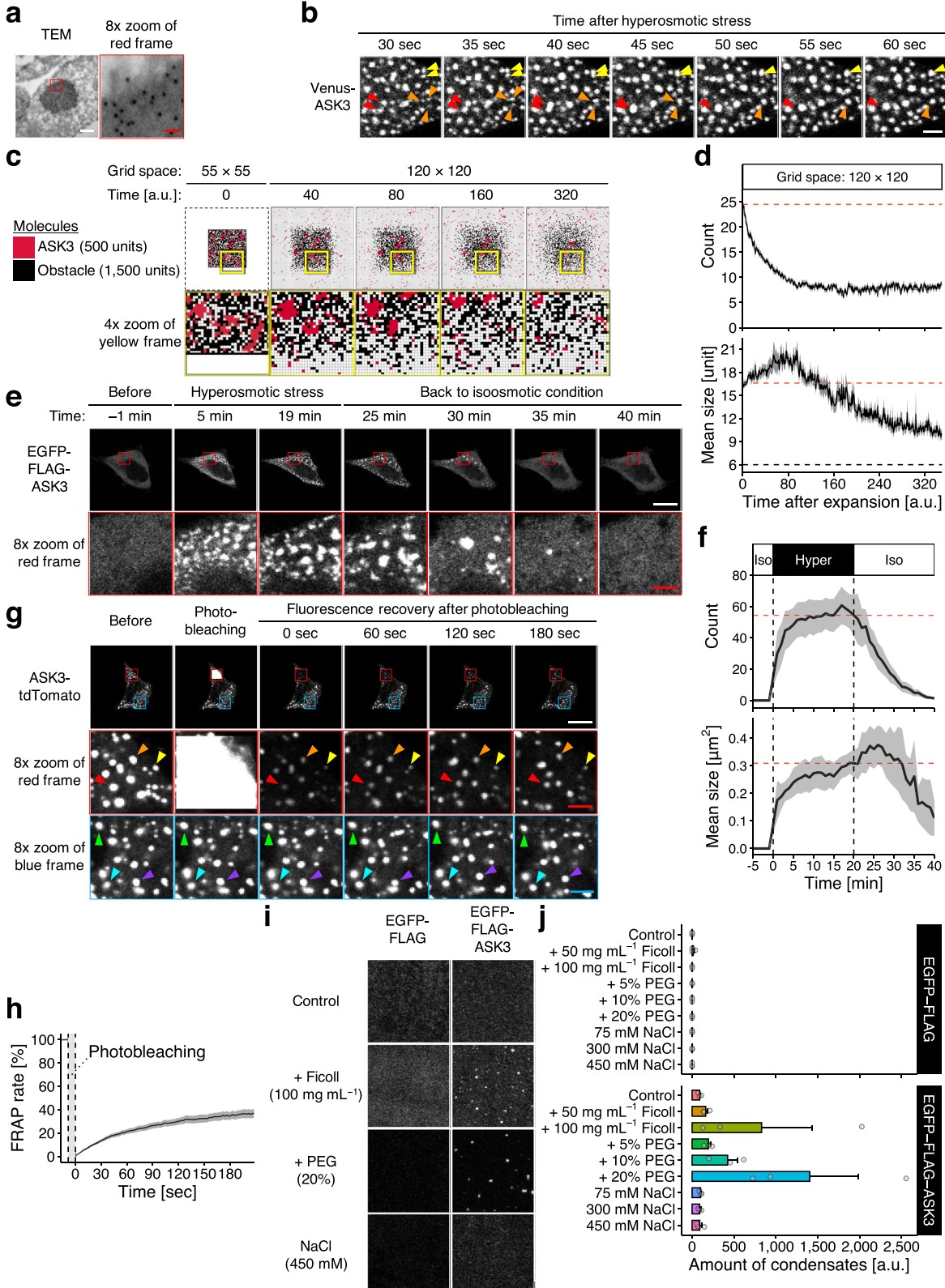

addition, we confirmed that full-length ASK3 mutant lacking both C-terminus CC (CCC) domain and LCR (CLCR) is unable to form condensates under hyperosmotic stress (Fig. 3c, Supplementary Fig. 3e, f).

Previously, we elucidated that ASK3 is dephosphorylated and inactivated a few minutes after hyperosmotic stress by protein phosphatase 6 (PP6)[18], implying that ASK3 condensation occurs prior to its inactivation. Besides, (1) ASK3 condensation was normally observed under hyperosmotic stress even when ASK3 inactivation was inhibited by the knockdown of PP6[18] (Supplementary Fig. 4a); (2) kinase-inactive ASK3 mutants K681M, which is mutated at the catalytic lysine in subdomain II of KD[17], and T808A, which is mutated at the phosphorylation site in activation loop of KD[17], did not form condensates under

**Fig. 2 ASK3 condensates are liquid-demixing condensates induced by liquid–liquid phase separation. a** Transmission electron microscopy (TEM) analysis with immunogold labeling for ASK3. Venus-ASK3-HEK293A cells were sampled after hyperosmotic stress (800 mOsm, 3 h). White bar: 250 nm, red bar: 31.25 nm. A representative image from ten condensates across five micrographs is presented. **b** Dynamics and fusion of ASK3 condensates in Venus-ASK3-HEK293A cells. Hyperosmotic stress: 500 mOsm, white bar: 2 μm. A representative image set from four independent experiments is presented. **c, d** A computational prediction for the number/size of ASK3 clusters after grid space expansion. Simulation results during $35 \times 10^6$ steps in rKMC method at the $120 \times 120$ grid space after the initial iteration with $5 \times 10^6$ steps at the $55 \times 55$ grid space are presented as the unit of time $t$ per $1 \times 10^5$ steps. Red dashed line: the mean of initial values at the grid space expansion, black dashed line: the minimum of ASK3 clusters definition. Data: mean ± SEM, $n = 12$ simulations. **e, f** Reversibility of ASK3 condensates in EGFP-FLAG-ASK3-transfected HEK293A cells. After hyperosmotic stress (600 mOsm, 20 min), the extracellular osmolality was set back to the isoosmotic condition. White bar: 20 μm, red bar: 2.5 μm. Red dashed line: the mean of initial values after setting back to isoosmotic condition, black dashed line: timepoint at the osmotic stress treatment. Data: mean ± SEM, $n = 8$ cells pooled from three independent experiments. **g, h** Fluorescence recovery after photobleaching (FRAP) assay for ASK3 condensates in ASK3-tdTomato-transfected HEK293A cells. Prior to the assay, cells were exposed to hyperosmotic stress (600 mOsm, 30 min). White bar: 20 μm, red/blue bar: 2.5 μm. Data: mean ± SEM, $n = 15$ cells pooled from five independent experiments. **i, j** ASK3 condensation in vitro. Control: 150 mM NaCl, 20 mM Tris (pH 7.5), 1 mM dithiothreitol (DTT), 15-min incubation on ice. Ficoll and polyethylene glycol (PEG): a crowding reagent, white bar: 5 μm. Data: mean ± SEM, $n = 3$ independent experiments.

isoosmotic conditions (Supplementary Fig. 4b); that is, it is suggested that ASK3 inactivation is neither necessary nor sufficient for the condensate formation of ASK3. Therefore, we evaluated the kinase activity of ASK3 mutants lacking condensation ability by utilizing the specific antibody for the phosphorylated Thr808 which is essential for the ASK3 kinase activity[17,25]. Although exhibiting lower basal activities under isoosmotic conditions, ASK3 ΔC and ΔCCCΔCLCR mutants were not dephosphorylated under hyperosmotic stress (Fig. 3d, Supplementary Fig. 5a), suggesting that ASK3 condensation is required for its inactivation. In fact, we discovered the unique relationship between ASK3 condensates and one of the PP6 subunits ANKRD52[18,26]; ANKRD52 condensates are not completely colocalized with ASK3 condensates, but they move around and grow while sharing their phase boundaries (Fig. 3e, Supplementary Movie 4).

**Nicotinamide phosphoribosyltransferase regulates PP6-mediated ASK3 inactivation via the NAD salvage pathway.** To reveal the details of ASK3 condensation and dephosphorylation, we investigated the candidate regulators of ASK3 inactivation identified by our genome-wide siRNA screen[18]. Among them, we focused on the highest-ranked and unexpected candidate nicotinamide phosphoribosyltransferase (NAMPT), the rate-limiting enzyme in the mammalian nicotinamide adenine dinucleotide (NAD) salvage pathway[27] (Fig. 4a, b). NAMPT knockdown inhibited ASK3 dephosphorylation under hyperosmotic stress (Fig. 4c, Supplementary Fig. 5b). In addition, NAMPT knockdown increased the phosphorylated endogenous ASK3 and decreased the phosphorylated STE20/SPS1-related proline/alanine-rich kinase (SPAK)/oxidative stress-responsive kinase 1 (OSR1) under hyperosmotic stress (Fig. 4d, Supplementary Fig. 5c), consistent with our previous finding that ASK3 downregulates SPAK/OSR1 in a kinase activity-dependent manner[17]. Overexpression of wild-type (WT) NAMPT fully accelerated ASK3 dephosphorylation under hyperosmotic stress in an amount-dependent manner (compare lanes 8–10 in Fig. 4e, Supplementary Fig. 5d). In accordance with the fact that NAMPT enzymatically functions as a homodimer[28], the homodimer-insufficient NAMPT mutant S199D promoted ASK3 dephosphorylation more slightly than WT NAMPT, and the homodimer-null NAMPT mutant S200D could not promote ASK3 dephosphorylation at all[29] (lanes 11–14 in Fig. 4e, Supplementary Fig. 5d). Furthermore, a NAMPT enzymatic inhibitor FK866[30] inhibited ASK3 dephosphorylation under hyperosmotic stress, which was canceled by further supplementation with the NAMPT enzymatic product nicotinamide mononucleotide (NMN) (Fig. 4f, Supplementary Fig. 5e). The knockdown of

cytosolic NMN adenylyltransferase NMNAT2[31,32] (Fig. 4b) inhibited ASK3 dephosphorylation under hyperosmotic stress (Supplementary Fig. 6a). These results indicate that not NAMPT itself but the subsequent reaction product NAD regulates ASK3 inactivation under hyperosmotic stress. As a molecular mechanism of ASK3 dephosphorylation under hyperosmotic stress, we previously revealed that not the phosphatase activity of PP6 but the interaction between PP6 and ASK3 is increased under hyperosmotic stress, followed by the direct dephosphorylation of ASK3 by PP6[18]. Pretreatment with FK866 and NMN reduced and promoted the interaction between PP6 and ASK3 under hyperosmotic stress, respectively (Fig. 4g, h, Supplementary Fig. 5f, g). Hence, NAMPT ensures ASK3 inactivation by regulating the PP6–ASK3 interaction under hyperosmotic stress.

**Poly(ADP-ribose) maintains the liquidity of ASK3 condensates for ASK3 inactivation.** NAD is not only a coenzyme in cellular redox reactions but also a substrate for enzymatic consumers, including cADP-ribose synthase CD38/157, sirtuins and PAR polymerases (PARPs)[33] (Fig. 4b). We thus examined the potential involvement of NAD-consuming enzymes in ASK3 inactivation. In contrast to the overexpression of NAMPT (Fig. 4e), neither CD38, SIRT2 nor PARP1 overexpression enhanced ASK3 dephosphorylation under hyperosmotic stress (Supplementary Fig. 6b–d); rather, their overexpression even inhibited ASK3 dephosphorylation, probably because they compete with the actual NAD-requiring regulators in ASK3 inactivation with respect to NAD. In the dynamics of poly(ADP-ribosyl)ation (PARsylation), however, not only PARPs (writers of PAR) but also the readers and erasers of PAR regulate PAR signaling[34–37] (Fig. 5a). Interestingly, when considering the involvement of NAD, we noticed that a PAR reader RING finger protein 146 (RNF146)[38–40] is a high-ranked candidate of ASK3 inactivator (Fig. 4a). Besides, NAMPT overexpression and FK866 pretreatment increased and decreased PARsylated proteins, respectively (Fig. 5b, Supplementary Fig. 5h). We therefore examined the potential involvement of PAR signaling in ASK3 inactivation by controlling a PAR eraser PAR glycohydrolase (PARG). PARG overexpression partially but significantly inhibited ASK3 dephosphorylation under hyperosmotic stress, while the glycohydrolase-inactive PARG mutant E673A/E674A[41] did not (Fig. 5c, Supplementary Fig. 5i), suggesting that PARsylation by unidentified PARP(s) or PAR per se is required for ASK3 inactivation under hyperosmotic stress.

PAR physicochemically resembles RNA. Similar to RNA, PAR is proposed to seed biomolecular condensates[42–45]. Considering that ASK3 condensation is required for its inactivation (Fig. 3),

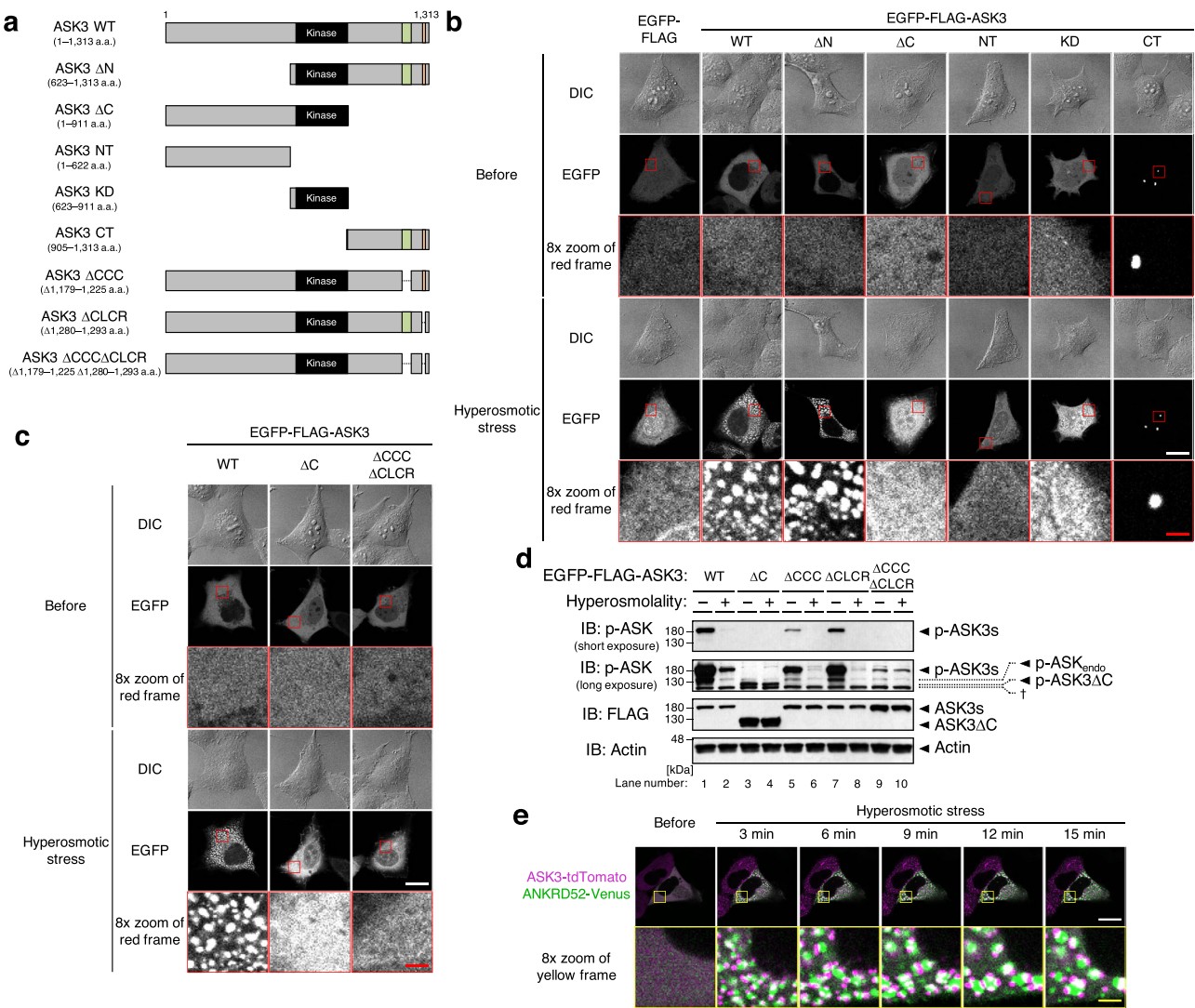

**Fig. 3 ASK3 condensation is required for ASK3 inactivation under hyperosmotic stress. a** Schematic representation of ASK3 deletion mutants and fragments. The numbers indicate the amino acid (a.a.) positions in wild-type (WT). Black rectangle: kinase domain (652–908 a.a.), green rectangle: C-terminus coiled-coil domain (CCC: 1179–1225 a.a.), orange rectangle: C-terminus low-complexity region (CLCR: 1280–1293 a.a.). **b, c** Subcellular localization of ASK3 deletion mutants and fragments in HEK293A cells. Hyperosmotic stress: 600 mOsm, 10 min. DIC: differential interference contrast, white bar: 20 μm, red bar: 2.5 μm. A representative image set from four independent experiments is presented. Note that the signal intensity of DIC cannot be compared among the images. **d** Requirement of CCC and CLCR for ASK3 inactivation in HEK293A cells. Hyperosmolality (−): 300 mOsm; (+): 500 mOsm; 10 min. IB: immunoblotting, p-ASK$_{endo}$: phosphorylation bands of endogenously expressed ASK. †Nonspecific bands. A representative image set from four independent experiments is presented (quantification: Supplementary Fig. 5a). **e** Relationship between ANKRD52 and ASK3 condensates in HEK293A cells. Magenta: ASK3-tdTomato, green: ANKRD52-Venus, hyperosmotic stress: 500 mOsm, white bar: 20 μm, yellow bar: 2.5 μm. A representative image set from five independent experiments is presented.

we next investigated the relationship between PAR and ASK3 condensation. Contrary to our expectation, ASK3 condensation under hyperosmotic stress was not prevented by the PAR depletion with FK866 pretreatment or PARG overexpression (Fig. 5d, e). However, FRAP of ASK3 condensates was significantly inhibited by the PAR depletion (Fig. 5f, g). Furthermore, PAR inhibited the formation of solid-like ASK3 condensates in vitro, while NAD could not (Fig. 5h, i). These results raise the possibility that PAR does not seed but rather "lubricates" phase-separated ASK3 for ASK3 inactivation. Of note, we found that poly(A) RNA also inhibits the formation of solid-like ASK3 condensates in vitro (Supplementary Fig. 7), suggesting that the common physicochemical property between PAR and RNA is important for the liquidity maintenance of ASK3 condensates.

To further explore the potential PAR-dependent conditioning of ASK3 condensates, we first generated ASK3 mutant that is basically insensitive to the PAR regulation. As a consensus sequence of the PAR-binding motif (PBM), $[HKR]_1-X_2-X_3-[AIQVY]_4-[KR]_5-[KR]_6-[AILV]_7-[FILPV]_8$ is proposed[46]. Based on the difference between WT ASK3 and its CT fragment in condensation at the basal state (Fig. 3b), some kind of inhibitory region is assumed to be present in the region before the ASK3 CT region, where the central positively charged $[KR]_5-[KR]_6$ is found in ten sites (named as PBM1–10, Fig. 6a). We thus constructed ten PBM candidate mutants substituted K/R with A. All PBM candidate mutants of ASK3 formed condensates under hyperosmotic stress (Fig. 6b). Between them, the ASK3 PBM4 mutant prominently exhibited the reduced dephosphorylation under hyperosmotic stress (Fig. 6c, Supplementary Fig. 5j). Indeed,

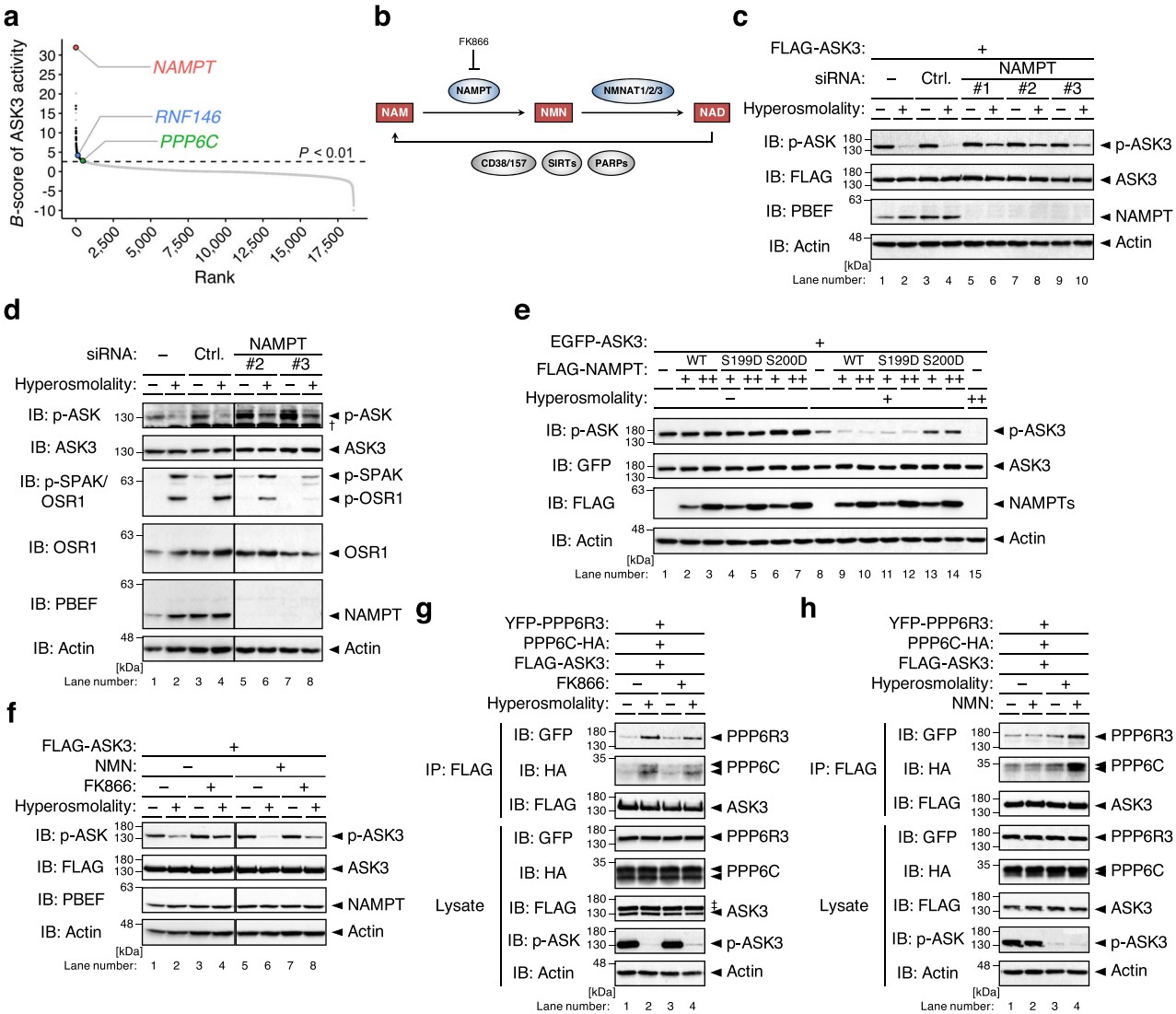

**Fig. 4 The NAD salvage pathway negatively regulates ASK3 activity under hyperosmotic stress. a** Distribution of gene candidates to regulate ASK3 inactivation in the previous primary screen[18]. A sample with a higher *B*-score corresponds to a higher potential candidate. PPP6C: the catalytic subunit of PP6, an ASK3 phosphatase. **b** Diagram of the mammalian nicotinamide adenine dinucleotide (NAD) salvage pathway. Rectangle: NAD-related molecule, arrow: reaction, ellipse: enzyme, NAM: nicotinamide, NMN: nicotinamide mononucleotide, FK866: a NAMPT enzymatic inhibitor[30]. **c** Effects of NAMPT depletion on ASK3 activity under hyperosmotic stress in FLAG-ASK3-stably expressing HEK293A (FLAG-ASK3-HEK293A) cells. **d** Effects of NAMPT depletion on endogenous ASK3 and SPAK/OSR1 activities under hyperosmotic stress in HEK293A cells. †Nonspecific bands. **e** Effects of NAMPT overexpression on ASK3 activity under hyperosmotic stress in HEK293A cells. WT: wild-type, S199D: homodimer-insufficient mutant, S200D: homodimer-null mutant[29]. **f** Effects of FK866 and/or NMN pretreatment on ASK3 activity under hyperosmotic stress in FLAG-ASK3-HEK293A cells. FK866 (−): dimethyl sulfoxide (DMSO), solvent for FK866; FK866 (+): 10 nM FK866; NMN (−): ultrapure water, solvent for NMN; NMN (+): 1 mM NMN; 3 h pretreatment. **g, h** Effects of FK866 or NMN pretreatment on the interaction between ASK3 and PP6 under hyperosmotic stress in HEK293A cells. FK866 (−): DMSO; FK866 (+): 10 nM FK866; NMN (−): ultrapure water; NMN (+): 1 mM NMN; 24 h pretreatment. ‡Remnant bands from prior detection of GFP. **c–h** Hyperosmolality (−): 300 mOsm; (+): 425 mOsm (with the exception in **g** and **h**, 500 mOsm); (++): 500 mOsm; 10 min. IB: immunoblotting, IP: immunoprecipitation. A representative image set from five (**c**, **f**) or four (**d**, **e**, **g**, **h**) independent experiments is presented (quantification: Supplementary Fig. 5b–g). Note that superfluous lanes were digitally eliminated from blot images in **d** and **f** as indicated by vertical black lines.

while WT ASK3 was coimmunoprecipitated with PAR by the pulldown of a PAR reader WWE domain[47] (Fig. 5a), the PBM4 mutant was not (Fig. 6d, Supplementary Fig. 5k), suggesting that ASK3 interacts with PAR via two arginine residues in the PBM4. In this WWE domain-utilized PAR pulldown assay, note that (1) the PAR depletion with FK866 pretreatment inhibited coimmunoprecipitation of WT ASK3, denying the possibility that ASK3 directly interacted with the WWE domain, and that (2) the PAR depletion with FK866 pretreatment did not significantly affect the slight remnant dephosphorylation of the PBM4 mutant under hyperosmotic stress (see the quantified data; Supplementary

Fig. 5j, k), suggesting that the effects of NAD on ASK3 inactivation is predominantly dependent on the PBM4. We subsequently investigated the material characteristics of the ASK3 PBM4 mutant condensates. Consistent with the PAR depletion (Fig. 5f, g), FRAP of the ASK3 PBM4 mutant condensates was significantly reduced although less drastically than that of the ASK3 CT fragment condensates (Fig. 6e, f). Moreover, the solid-like condensates of the ASK3 PBM4 mutant in vitro remained even under the PAR addition (Fig. 6g, h). These findings reinforce the notion that PAR keeps ASK3 condensates in the liquid phase, enabling ASK3 to be inactivated under hyperosmotic stress.

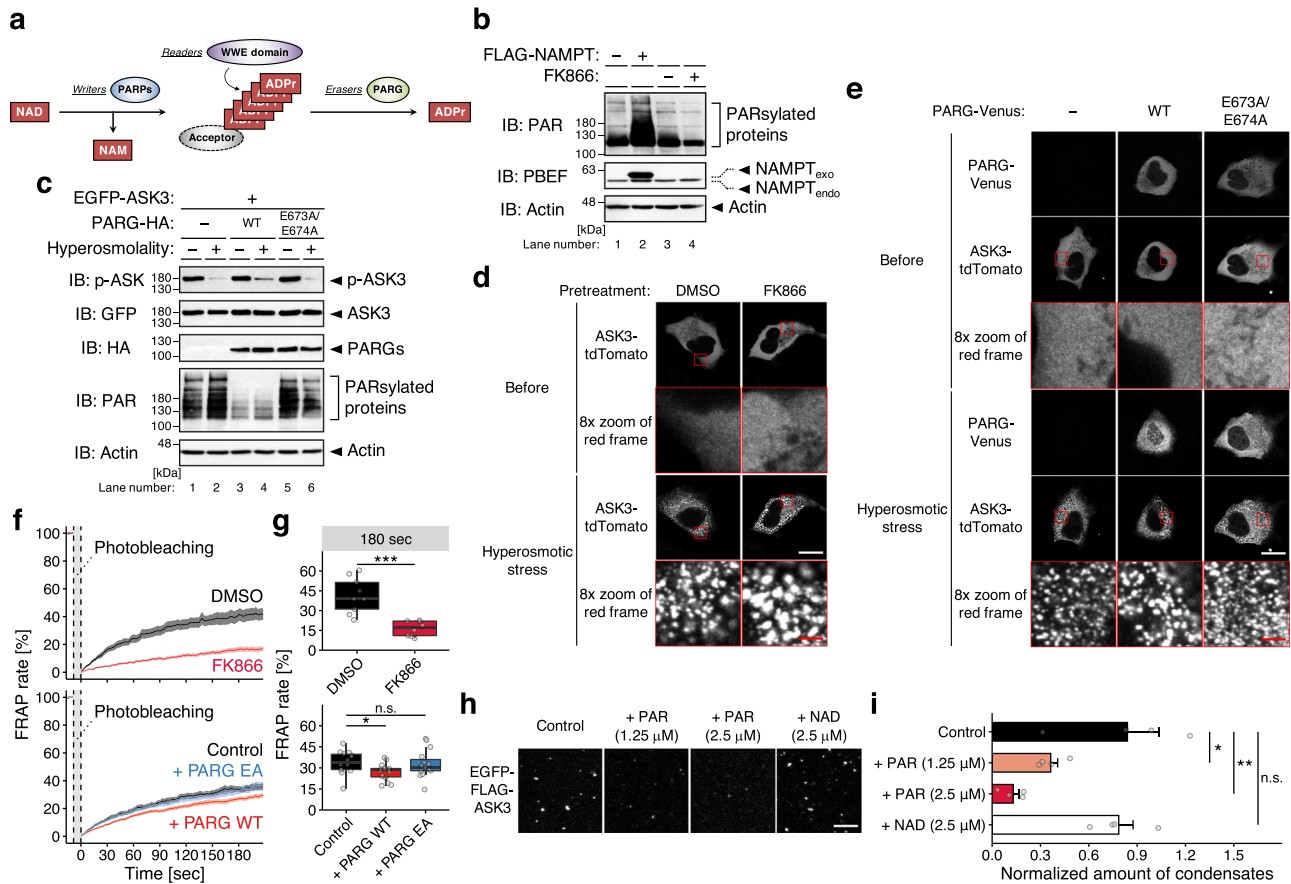

**Fig. 5 Poly(ADP-ribose) keeps ASK3 condensates in the liquid phase for ASK3 inactivation. a** Diagram of the poly(ADP-ribosyl)ation (PARsylation) dynamics. Rectangle: NAD-related molecule, arrow: reaction, ellipse: enzyme, ADPr: ADP-ribose. **b** Effects of NAMPT overexpression or inhibition on the amount of PAR in HEK293A cells. FK866 (−): dimethyl sulfoxide (DMSO), solvent for FK866; (+): 10 nM FK866; 18–24 h pretreatment. NAMPT$_{exo}$: exogenously expressed NAMPT, NAMPT$_{endo}$: endogenously expressed NAMPT. **c** Effects of PARG overexpression on ASK3 activity under hyperosmotic stress in HEK293A cells. WT: wild-type, E673A/E674A: glycohydrolase-inactive mutant[41]. Hyperosmolality (−): 300 mOsm; (+): 425 mOsm; 10 min. **b**, **c** IB: immunoblotting. A representative image set from three (**b**) or four (**c**) independent experiments is presented (quantification: Supplementary Fig. 5h, i). **d**, **e** Effects of PAR depletion on ASK3 condensates in ASK3-tdTomato-transfected HEK293A cells. Hyperosmotic stress: 600 mOsm, 10 min. White bar: 20 µm, red bar: 2.5 µm. A representative image set from four independent experiments is presented. **f**, **g** Effects of PAR depletion on the FRAP of ASK3 condensates in ASK3-tdTomato-transfected HEK293A cells. Prior to the assay, cells were exposed to hyperosmotic stress (600 mOsm, 30 min). Data (**f**): mean ± SEM; data (**g**): center line = median; box limits = [$Q_1$, $Q_3$]; whiskers = [max(minimum value, $Q_1$ − 1.5 × IQR), min(maximum value, $Q_3$ + 1.5 × IQR)], where $Q_1$, $Q_3$ and IQR are the first quartile, the third quartile and the interquartile range, respectively; $n$ = 8 (FK866), 9 (DMSO), 14 (Control), 15 (+PARG WT and EA) cells pooled from three (top panels) or five (bottom panels) independent experiments. *$P < 0.05$, ***$P < 0.001$, n.s. (not significant) according to two-sided Welch's $t$-tests (with the Bonferroni correction in the bottom panel). **d**–**g** DMSO: solvent for FK866; FK866: 10 nM FK866; 18–24 h pretreatment. Control: empty vector, PARG WT: wild-type PARG-Venus, PARG EA: E673A/E674A mutant PARG-Venus. **h**, **i** Effects of PAR on solid-like ASK3 condensation in vitro. Control: 150 mM NaCl, 20 mM Tris (pH 7.5), 1 mM DTT, 20% PEG, 15-min incubation on ice. White bar: 5 µm. Data: mean ± SEM, $n$ = 4 independent experiments. *$P < 0.05$, **$P < 0.01$, n.s. (not significant) according to two-sided Dunnett's test.

## Discussion

In general, cells face three types of perturbations after osmotic stress: changes in mechanical forces in/on the phospholipid bilayer, intracellular strength/concentration of ions and macro-molecular crowding[8]. It has been suggested that cells recognize unsubstantial osmotic stress through these changes. Here, we demonstrated that an osmoresponsive kinase ASK3 quickly forms liquid-demixing condensates after hyperosmotic stress, followed by the regulation of its kinase activity. Our computational model and in vitro assays suggested that the change in macromolecular crowding is a driving force for the condensation. Unlike recent studies on potential osmosensors in/on the cell membrane, our findings shed light on another mechanism that cells sense osmotic stress from the inside through LLPS (Fig. 7a). In fact, when applying the paradigm of LLPS, we can reassess that another osmoresponsive kinase WNK1 also forms liquid-demixing

condensates albeit only regarded as WNK puncta or WNK bodies thus far[48–50]. In addition to the osmoresponsive kinases in cell volume regulation, the biomolecular condensates induced by LLPS just seconds after hyperosmotic stress have been reported very recently[51,52]. Our findings are reflective of a theory of cellular volume sensing put forth by Zimmerman and Harrison in 1987: cells sense changes in their volume simply through changes in reaction rates induced by changes in macromolecular crowding[53]. We can advance upon and generalize their idea: cells sense changes in their volume through phase separation/transition triggered by changes in macromolecular crowding.

Recent studies on LLPS have gradually unveiled the versatile functions of biomolecular condensates, including the acceleration or suppression of specific reactions, the buffering effects on specific biomolecule concentrations, and even the selective filters of nuclear pores[12,13,15]. In this study, we discovered the dual

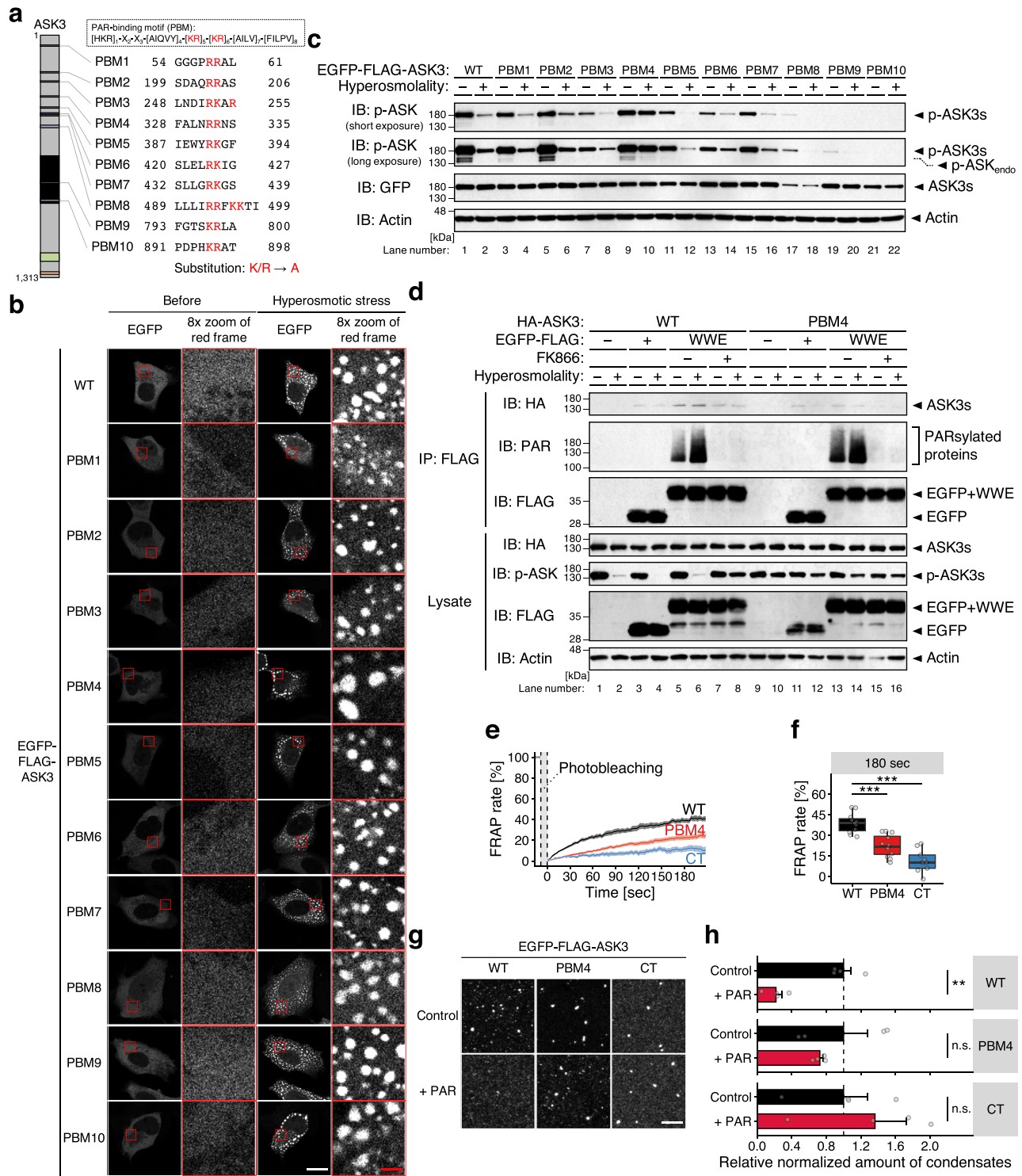

function of ASK3 condensates. One function involves the sensing machinery for osmotic stress. Stress-sensing condensates were also suggested in yeast under thermal or pH stresses[54]. Interestingly, all of these stresses are bidirectionally induced by deviation from steady state; thus, the role of liquid-demixing droplets would be rational in quick and reversible stress recognition. The other is the dephosphorylation/inactivation of ASK3. Because the dephosphorylation competes with the autophosphorylation of ASK3 at the basal level, this would be interpreted as not the trigger of reaction but the acceleration of reaction specificity. As another possibility, however, ASK3 condensates may serve as

multifunctional signaling hubs for the whole regulation of ASK3 activity more generally. In fact, the ASK3 ΔC and ΔCCCΔCLCR mutants, which are unable to form condensates (Fig. 3b, c), exhibited lower phosphorylated level even under basal conditions (Fig. 3d). Although our confocal microscopy detected few ASK3 condensates under isoosmotic conditions (Fig. 1a), our computational simulation indicated that small clusters appear stochastically and transiently under larger grid space (Fig. 1d, Supplementary Movie 1), which may be related to the reports on local phase separation[55] and nanometer-sized condensates[56]. This notion that ASK3 condensates provide condition-dependent

**Fig. 6 Interaction between ASK3 and PAR is required for the liquidity of ASK3 condensates and ASK3 inactivation under hyperosmotic stress.**
**a** Candidate PAR-binding motif (PBM) in ASK3. Schematic representation of ASK3 is the same as that in Fig. 3a. The numbers indicate the amino acid (a.a.) positions in wild-type (WT). **b** Subcellular localization of ASK3 PBM candidate mutants in HEK293A cells. Hyperosmotic stress: 500 mOsm, 10 min. White bar: 20 μm, red bar: 2.5 μm. A representative image set from three independent experiments is presented. **c** Inactivation of ASK3 PBM candidate mutants under hyperosmotic stress in HEK293A cells. p-ASK$_{endo}$: phosphorylation bands of endogenously expressed ASK. **d** Ability of the ASK3 PBM4 mutant to interact with PAR under hyperosmotic stress in HEK293A cells. FK866 (−): dimethyl sulfoxide (DMSO), solvent for FK866; (+): 10 nM FK866; 12 h pretreatment. **c, d** Hyperosmolality (−): 300 mOsm; (+): 500 mOsm; 10 min. IB: immunoblotting, IP: immunoprecipitation. A representative image set from six (**c**) or four (**d**) independent experiments is presented (quantification: Supplementary Fig. 5j, k). **e, f** FRAP assay for condensates of the ASK3 PBM4 mutant in HEK293A cells. Prior to the assay, cells transfected with each ASK3-tdTomato were exposed to hyperosmotic stress (600 mOsm, 30 min). Data (**e**): mean ± SEM; data (**f**): center line = median; box limits = $[Q_1, Q_3]$; whiskers = $[\max(\text{minimum value}, Q_1 - 1.5 \times \text{IQR}), \min(\text{maximum value}, Q_3 + 1.5 \times \text{IQR})]$, where $Q_1$, $Q_3$ and IQR are the first quartile, the third quartile and the interquartile range, respectively; $n = 9$ (CT), 11 (WT), 12 (PBM4) cells pooled from four independent experiments. ***$P < 0.001$ according to two-sided Welch's $t$-tests with the Bonferroni correction. **g, h** Effects of PAR on the solid-like condensates of the ASK3 PBM4 mutant in vitro. Control: 150 mM NaCl, 20 mM Tris (pH 7.5), 1 mM DTT, 20% PEG, 15-min incubation on ice. PAR: 2.5 μM, white bar: 5 μm. Data: mean ± SEM, $n = 4$ independent experiments. *$P < 0.05$, n.s. (not significant) according to two-sided Student's $t$-test with the Bonferroni correction.

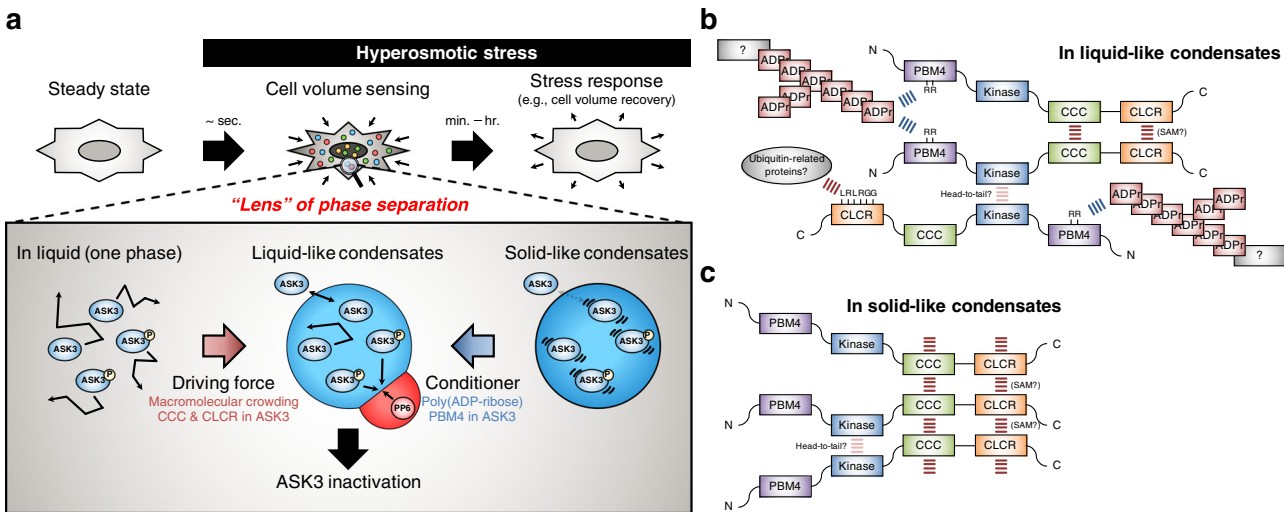

**Fig. 7 Schematic summary of the main findings and molecular mechanism models.** **a** Schematic representation of the main findings in this study. Under hyperosmotic stress, cells sense cell shrinkage immediately and induce appropriate responses to maintain homeostasis. By leveraging the "lens" of phase separation, we unveiled that cells rationally incorporate macromolecular crowding-driven phase separation into their osmosensing systems. In addition to the sensing machinery for osmotic stress, ASK3 condensates function as the specific accelerator of ASK3 inactivation under hyperosmotic stress (at least). Unlike the roles in condensates of most RNA-binding proteins, poly(ADP-ribose) (PAR) endows ASK3 condensates with the liquid property and enables the PP6-mediated inactivation of ASK3. **b, c** A current schematic model for the molecular mechanism of ASK3 condensates under hyperosmotic stress. Under the excluded volume effect, the heterogenous multivalent modular interactions via C-terminus coiled-coil (CCC) domain and low-complexity region (CLCR) would drive the condensate formation of ASK3 (indicated as red bars). At the same time, the flexible and bulky PAR would also multivalently interact with ASK3 via arginine residues in the PAR-binding motif (PBM4) of ASK3 and indirectly coordinate the multivalent modular interactions between ASK3 molecules, which would control the liquidity of ASK3 condensates (indicated as blue bars in **b**). When the multivalent interactions between ASK3 and PAR are absent, the modular interactions between ASK3 molecules would be strong enough to drive the maturation of ASK3 condensates into the solid phase (**c**).

signaling hubs may resolve the question why the autophosphorylation/activation of ASK3 is not predominant within the dense condensates under hyperosmotic stress. To explore this possibility, further analyses, such as the identification of biomolecules within ASK3 condensates, are needed.

Multivalent interactions are key molecular driving forces for the phase separation of biomolecular condensates[12–15]. In particular, IDR, which exhibits flexible conformations and enables weak transient interactions, is required for the phase separation of many RNA-binding proteins (RBPs). Although IDRs are predicted in ASK3, the CC domain and LCR in ASK3 CT region were essential for the ASK3 condensation (Fig. 3c, Supplementary Fig. 3). CC domain is a famous oligomerization domain that composes a supercoil of more than two α-helixes and contributes to the formation of canonical protein complex[57]; thus, CC

domain may seem to generate too strong interaction for phase separation, but the CC domain-driven phase separation is suggested in other biomolecular condensates[58]. The CLCR of ASK3 is localized within SAM domain (Supplementary Fig. 3a). SAM domain is also known as a conventional interaction module and the very recent paper revealed that the SAM domain of ASK3 forms a stable homooligomer in vitro[24]. Although our results using LCR deletion may reflect on the disruption of functional SAM domain, CLCR of ASK3 has not only the enriched residues of arginine, which would endow the electrostatic or cation-π interactions observed in the IDR of many biomolecular condensates, but also the ubiquitin CT sequence LRLRGG, which may be interacted with various ubiquitin-related proteins[59,60]. Hence, the heterogenous multivalent modular interactions would govern the condensate formation of ASK3 (Fig. 7b).

At the same time, we found that PAR interacts with ASK3 at least via two arginine residues in the PBM4 and is required for the liquidity of ASK3 condensates (Figs. 5, 6). PAR consists of three components—a nucleobase (adenine), ribose and phosphate—which are the same as RNA. In most RBP condensates, RNA promotes LLPS partly because the multivalent interactions between RNA and RBPs reduce the threshold for LLPS[12–15]. Likewise, PAR has been reported to promote condensation[43–45]. Due to flexible and branching structures, however, PAR is assumed to contribute not only multivalent interactions but also condensate porosity, which significantly affects the viscoelasticity of biomolecular condensates[12,14]. In case of ASK3 condensates, the combination of CCC and CLCR additively or synergistically generates strong interactions between ASK3 molecules based on the results that the ASK3 CT fragment forms the solid-like condensates even under isoosmotic conditions (Figs. 3b, 6e, f) and that the singly deleted mutants CTΔCC and CTΔLCR still possesses the condensation ability in response to hyperosmotic stress (Supplementary Fig. 3c, d). Moreover, considering that ASK3 KD fragment itself has the insufficient ability of condensation in cells (Fig. 3b) but purified KD of ASK1 which is highly homologous to ASK3 KD[17] tightly dimerizes in vitro[61], KD of ASK3 may cause further sticky interaction between ASK3 molecules only following the interactions via CCC and CLCR. A single interaction between PAR and ASK3 is weaker than these tight potential interactions between ASK3 molecules, but the weak multivalent interactions of PAR would trap ASK3 transiently and mitigate the overall tight interactions between ASK3 molecules, improving the liquidity of ASK3 condensates. In addition, the bulky PAR can generate free space within the condensates and increase the ASK3 movement, which would enable ASK3 to approach PP6. Therefore, while indirectly coordinating the multivalent modular interactions between ASK3 molecules, PAR would keep the liquidity of ASK3 condensates within the "Goldilocks zone" and provide opportunity for the transient interaction between ASK3 and PP6 around the phase boundaries, followed by ASK3 inactivation (Fig. 7a, b); otherwise, ASK3 condensates undergo liquid-to-solid transition (Fig. 7c), analogous to RNA buffering for prion-like RBPs[62]. This interpretation accords with our results: (1) the PAR depletion with FK866 pretreatment (Fig. 5b) reduced the interaction between PP6 and ASK3 under hyperosmotic stress (Fig. 4g); (2) PP6 condensates were not completely incorporated in ASK3 condensates but partly shared the phase boundary with ASK3 condensates (Fig. 3e), demanding that the dephosphorylated ASK3 around the phase boundary must be smoothly exchanged for the still phosphorylated ASK3 to maintain the efficient dephosphorylation by PP6 within the restricted space; (3) condensates of the PAR-unbinding ASK3 PBM4 mutant (Fig. 6) and the WT ASK3 under PAR depletion with FK866 pretreatment (Fig. 5b) merged with PP6 condensates but instead lost the shared phase boundaries (Supplementary Fig. 8). Note that the above points (1) and (3) may seem to be contradictory, but the colocalization observed in conventional fluorescence microscopy does not necessarily indicate the physical molecule–molecule interaction because we can only detect the expanded fluorescent signals as a single sub-micrometer dot. Therefore, as one of the coherent explanations for the seemingly contradictory findings, we hypothesize that the functional physical interactions between PP6 and ASK3 occur only in the surface vicinity of condensates and that the PP6–ASK3 interactions within the core of condensates do not occur or occur just as pseudo stochastic interaction. Rationality of the strategy to use the surface vicinity of condensates in ASK3 dephosphorylation remains to be clarified, but this strategy may be advantageous according to the results that (1) the number and size of ASK3 condensates increases and decreases, respectively,

under stronger hyperosmotic stress (Fig. 1b), indicating that the cellular total surface area of condensates increases in a hyperosmolality strength-dependent manner, and that (2) ASK3 dephosphorylation is enhanced in a hyperosmolality strength-dependent manner[17]. Since the existence of the partly contacted condensates of PP6 and ASK3 suggests that the surface tensions from three interfaces, ASK3–cytosol, cytosol–PP6 and PP6–ASK3, are balanced[13], the PP6–ASK3 interface may be evolved to function as the private channel for PP6 and ASK3 within the limited space under hyperosmotic stress. Of note, the liquid-to-solid transition of ASK3 condensates may also be a cellular functional response in a certain environment. We previously discovered that the insufficient inactivation of ASK3 under hyperosmotic stress leads to the failure of cell volume recovery, followed by the cell death[18]. In a regulated necrotic cell death parthanatos, PAR is accumulated by PARP1 overactivation and triggers the translocation of apoptosis-inducing factor[63]. Considering that PARP1 overexpression inhibited ASK3 dephosphorylation under hyperosmotic stress (Supplementary Fig. 6d), therefore, the regulation of PAR dynamics may efficiently link the phase regulation of ASK3 condensates with the cell death induction.

One of the largest barriers in this study is the situation that we have not identified the responsible PARP(s) in ASK3 inactivation thus far. Among 17 members of the human PARP family, PARP1, PARP2, TNKS and TNKS2 are currently said to have the ability of PARsylation[35,36,64]; especially, TNKS and TNKS2 are primarily localized in cytosol and have the SAM domain for oligomerization[65], having raised our anticipation. However, the overexpression of TNKS or TNKS2 inhibited ASK3 dephosphorylation under hyperosmotic stress in a PARP activity-dependent manner[65] (Supplementary Fig. 6e, f). Besides the result that PARP1 overexpression inhibited ASK3 dephosphorylation under hyperosmotic stress (Supplementary Fig. 6d), a PARP inhibitor 1,5-dihydroxyisoquinoline whose broad spectrum covers PARP1 and PARP2[66] enhanced ASK3 dephosphorylation under hyperosmotic stress (Supplementary Fig. 6g). Furthermore, our genome-wide siRNA screen[18] identified no known PARP family member as an ASK3 inactivator candidate. Based on the potential that PAR can be synthesized and degraded within seconds[64], the responsible PARP(s) in ASK3 inactivation may increase the amount of PAR in response to hyperosmotic stress, raising the possibility that PAR inhibits the liquid-to-solid transition of ASK3 condensates after the initial LLPS. Or more simply, the basal level of PAR may be sufficient for the liquidity of ASK3 condensates, implying the mechanism that PAR adjusts the depth of phase separation from the beginning of LLPS. In another point, PARsylation is a posttranslational modification and PAR is usually conjugated with proteins, while the PAR erasers can generate free PAR[37]. Because PARG has both exoglycosidic and endoglycosidic activities to release free ADP-ribose and free PAR, respectively, the results of PARG overexpression (Fig. 5c, e–g) do not deny the possibility that free PARs regulate ASK3 condensates. As discussed in the recent review[64], the further analyses for the site, length, and structure of PAR would develop our understanding of PAR regulation in biomolecular condensates.

## Methods

**Materials**. Reagents, expression plasmids, siRNAs and protein purification are described in Supplementary Methods[67,68], and the key materials used in this study are summarized in Supplementary Data 1.

**Cell culture**. HEK293A cells (Invitrogen) were cultured in Dulbecco's modified Eagle's medium (DMEM) (Sigma-Aldrich, Cat. #D5796) supplemented with 10% fetal bovine serum (FBS; BioWest, Cat. #S1560-500) and 100 units mL$^{-1}$ penicillin G (Meiji Seika, Cat. #6111400D2039). Tetracycline-inducible Venus-ASK3-stably expressing HEK293A (Venus-ASK3-HEK293A) cells were established with the

T-REx system (Invitrogen). Tetracycline-inducible FLAG-ASK3-stably expressing HEK293A (FLAG-ASK3-HEK293A) cells were established previously[18]. Venus-ASK3-HEK293A cells and FLAG-ASK3-HEK293A cells were cultured in DMEM supplemented with 10% FBS, 2.5 µg mL$^{-1}$ blasticidin (Invitrogen, Cat. #A1113903) and 50 µg mL$^{-1}$ Zeocin (Invitrogen, Cat. #R25001). To induce Venus-ASK3 or FLAG-ASK3, the cells were pretreated with 1 µg mL$^{-1}$ tetracycline (Sigma-Aldrich, Cat. #T7660) 24 h before assays. All cells were cultured in 5% $CO_2$ at 37 °C and verified to be negative for mycoplasma.

**Transfection**. Plasmid transfections were performed with polyethylenimine "MAX" (Polysciences, Cat. #24765) when HEK293A cells were grown to 95% confluency, according to the same protocol with the previous study[18]. To reduce cytotoxicity, the cells were cultured in fresh medium 6–10 h later, followed by another 40 h of culture. siRNA transfections for FLAG-ASK3-HEK293A cells were carried out by forward transfection using Lipofectamine RNAiMAX (Invitrogen, Cat. #133778-500) and siRNAs (10 nM for Stealth siRNAs, 30 nM for siGENOME siRNAs) once the cells reached 40–80% confluency, according to the manufacturer's instructions. siRNA transfections for HEK293A cells were carried out by reverse transfection using Lipofectamine RNAiMAX and 30 nM siRNAs, according to the manufacturer's instructions.

**Osmotic stress treatment**. In live-cell imaging experiments, osmotic stress was applied by adding the 2× osmotic medium into the culture medium, followed by the incubation in 5% $CO_2$ at 37 °C. For isoosmotic conditions (~300 mOsm (kg $H_2O$)$^{-1}$), DMEM supplemented with 10% FBS was used as the isoosmotic medium. For hyperosmotic stress (~400, ~500, ~600 or ~700 mOsm (kg $H_2O$)$^{-1}$), DMEM supplemented with 10% FBS and 200, 400, 600 or 800 mM mannitol was used as the 2× hyperosmotic medium. In the case of NaCl-based hyperosmotic stress (~400, ~500 or ~600 mOsm (kg $H_2O$)$^{-1}$), DMEM supplemented with 10% FBS and 100, 200 or 300 mM NaCl was used as the 2× hyperosmotic medium. For hypoosmotic stress (~150 or 225 mOsm (kg $H_2O$)$^{-1}$), ultrapure water or twofold diluted isoosmotic medium was used as the 2× hypoosmotic medium.

In immunoblotting experiments, osmotic stress was applied by exchanging the culture medium with osmotic buffer. The isoosmotic buffer (300 mOsm (kg $H_2O$)$^{-1}$, pH 7.4) contained 130 mM NaCl, 2 mM KCl, 1 mM $KH_2PO_4$, 2 mM $CaCl_2$, 2 mM $MgCl_2$, 10 mM 4-(2-hydroxyethyl)-1-piperazineethanesulfonic acid, 10 mM glucose and 20 mM mannitol. The hyperosmotic buffer (425 or 500 mOsm (kg $H_2O$)$^{-1}$, pH 7.4) was the same as the isoosmotic buffer but contained 145 or 220 mM mannitol, respectively. Absolute osmolality was verified by an Osmomat 030 (Gonotec) osmometer to fall within the range of 295 to 320 mOsm (kg $H_2O$)$^{-1}$ for isoosmotic buffer or ±25 mOsm (kg $H_2O$)$^{-1}$ for the other buffers.

**Live-cell imaging**. Cells were seeded in 35 mmφ glass bottom dishes (Matsunami, Cat. #D11130H) which were coated with 1% gelatin (Nacalai Tesque, Cat. #16605-42) in phosphate-buffered saline (PBS; 137 mM NaCl, 3 mM KCl, 8 mM $Na_3PO_4$•12$H_2O$, 15 mM $KH_2PO_4$) in advance. For transfected HEK293A cells, the cells were reseeded from a 24-well plate into the glass bottom dishes 24 h after transfection. After 36–60 h, the culture medium was replaced with 1 mL isoosmotic medium per dish, and the dish was subsequently viewed by a TCS SP5 (Leica) confocal laser-scanning microscope equipped with a stage top incubator (Tokai Hit). The cells were observed in 5% $CO_2$ at 37 °C using an HC PL APO 63×/1.40 oil objective (Leica). Multichannel time-lapse images were acquired in four fields each with four averages per frame in 1- or 1.5-min intervals. Venus, EGFP or tdTomato was excited at 514 nm with an argon laser, at 488 nm with an argon laser or at 561 nm with a DPSS laser, respectively, and detected by a HyD detector (Leica). Differential interference contrast (DIC) was captured through the transmitted light from either the argon or DPSS laser, the unused laser in the observation, and detected by a PMT detector (Leica). After obtaining image sets for 5 min under isoosmotic conditions as the "Before" condition, the cells were exposed to osmotic stress by adding 1 mL of 2× osmotic medium per dish and continuously observed for 30 min. Of note, although the cellular morphology was appreciably changed under osmotic stress, we observed the constant position, XY and focal plane, by utilizing a motorized stage and the on-demand mode of adaptive focus control system (Leica) in each field.

In the experiments of the dynamics and fusion of ASK3 condensates, single-channel time-lapse imaging for Venus was performed in a single field with two averages per frame at the minimum interval (~1 s). To hold a constant position and minimize the autofocusing time, the continuous mode of adaptive focus control system was applied. Similarly, the relationship between ANKRD52 and ASK3 condensates was investigated by 2-channel time-lapse imaging for Venus and tdTomato at the minimum interval (~5 s).

In the experiments of the reversibility of ASK3 condensates, time-lapse images of the EGFP and DIC channels were captured by the following procedure. After acquiring the "Before" image sets for 5 min under isoosmotic conditions, the cells were exposed to hyperosmotic stress (600 mOsm) by adding 1 mL of 2× hyperosmotic medium per dish and observed for 20 min. Subsequently, the cells were reverted back to isoosmotic conditions by adding 2 mL of ultrapure water per dish and further observed for 20 min.

For presentation, representative raw images were adjusted in brightness and contrast linearly and equally within the samples by using the GNU Image Manipulation Program (GIMP; GIMP Development Team, https://www.gimp.org/) or Fiji/ImageJ[69] (https://fiji.sc/) software. Because we used DIC images as a rough confirmation of cytosol region, automatically optimized adjustment in brightness and contrast was applied to each DIC image; therefore, the signal intensity of DIC image cannot be compared among the images. To create a time-lapse video, a series of images were equally adjusted in brightness and contrast (and assigned colors if multiple channels were included), captions were added, and the images were converted to a movie file using Fiji/ImageJ software.

For quantification, we established a macro script in Fiji/ImageJ to calculate the count and size of ASK3 condensates in a cell per frame and applied it to all raw image sets in batch mode. Briefly, based on a DIC image, the region of interest (ROI) was first defined as the whole cell area of a main cell because there are condensates from another cell in some cases. After applying a Gaussian filter, the Venus signal within the ROI was subsequently extracted from a Venus image in accordance with the local threshold. Finally, particle analysis was performed. Each parameter was determined from pilot analyses in Venus-ASK3-HEK293A cells. The exported data table was summarized with R language on RStudio (RStudio, Inc., https://rstudio.com/) software. The Fiji/ImageJ script also exported images of both ROIs and identified particles, enabling us to confirm the quality. In fact, we excluded several data points from the data analysis: (1) if the image was out-of-focus, (2) if the target cell was shrunken too much or detached completely or (3) if the value was an extreme outlier, less than $Q_1 - 5 \times IQR$ or more than $Q_3 + 5 \times IQR$, where $Q_1$, $Q_3$ and IQR are the first quartile, the third quartile and the interquartile range, respectively.

**Fluorescence recovery after photobleaching assay**. Ideally, FRAP should be applied to only a single condensate. However, ASK3 condensates move around too dynamically and rapidly to be evaluated by normal FRAP assay; ASK3 condensates go out from the focal plane and vice versa, for example. Hence, we established and performed a subsequent FRAP assay for ASK3 condensates under hyperosmotic stress. Prior to the FRAP assay, ASK3-tdTomato-transfected HEK293A cells were placed under the microscope in 5% $CO_2$ at 37 °C and exposed to hyperosmotic stress (600 mOsm) for 30 min, which makes ASK3 condensates relatively stable. Subsequently, single-channel time-lapse imaging for tdTomato was performed in a single field with four averages per frame with a minimum interval (~2.1 s) by utilizing the continuous mode of adaptive focus control system. After acquiring five frames as the "Before" condition, a rectangular area that included more than ten condensates (with the exception of the ASK3 CT mutant for which one condensate was included) was photobleached by the maximum intensity of the DPSS laser three times, followed by the time-lapse imaging of 100 intervals as the "After" condition.

To quantify the FRAP rate of ASK3 condensates from image data, particle tracking analysis was first executed for all ASK3 condensates by using a Fiji plugin TrackMate[70] (https://imagej.net/TrackMate). In TrackMate, each condensate was identified in each frame by a Laplacian of Gaussian detector, followed by connecting frames by a linear assignment problem tracker. Each parameter was determined from pilot analyses for ASK3(WT)-tdTomato. In this tracking analysis, we excluded the condensates (1) that were not successfully tracked from "Before" to "After" or (2) that were present in less than 25 frames. Next, the tracking data table was systematically calculated to the FRAP rate in RStudio software. In the R script, each tracked condensate was first categorized into two groups, photobleached or not-photobleached, based on the XY coordinates of the photobleached rectangular area. Meanwhile, the fluorescence intensity value of condensate $i$ at time $t$, $F_i(t)$ was converted to the relative fluorescence change $F_i(t)/F_{i,Before}$, where $F_{i,Before}$ indicates the mean of $F_i(t)$ for each condensate $i$ in the "Before" condition. At this step, we eliminated a few false positive condensates in the photobleached group whose $F_i(t)/F_{i,Before}$ did not exhibit at least a 15% decrease between the "Before" and "After" conditions, although $F_i(t)/F_{i,Before}$ of the other photobleached condensates dropped by an average of 80% in our FRAP assays. To correct the quenching effects during observation, each $F_i(t)/F_{i,Before}$ in the photobleached group was normalized to $G_i(t) = (F_i(t)/F_{i,Before})/$(the mean of $F_j(t)/F_{j,Before}$ in the nonphotobleached group). To mitigate the effects of condensate movement in a direction vertical to the focal plane on the changes in fluorescence, $G_i(t)$ was further converted to the mean of $G_i(t)$ in the photobleached group, $G(t)$; namely, we summarized all values of the photobleached condensates in a cell into representative values of one virtual condensate. Finally, the FRAP rate [%] at time $t$ in the cell was calculated as $(G(t) - G_{Min})/(1 - G_{Min}) \times 100$, where $G_{Min}$ was the minimum value within the first three time points of the "After" condition. When summarizing the FRAP rate between cells, we trimmed a few extreme outliers, less than $Q_1 - 5 \times IQR$ or more than $Q_3 + 5 \times IQR$.

**Immunocytochemistry and immunofluorescence**. Transfected HEK293A cells were seeded on 15 mmφ cover slips (Matsunami, Cat. #C015001) in a 12-well plate which were coated with 1% gelatin in PBS in advance. After 24–48 h, the cells were exposed to osmotic medium or buffer for the indicated period, followed by the following immunostaining steps: fixation for 15 min with 4% formaldehyde (Wako Pure Chemical Industries, Cat. #064-00406) in PBS, permeabilization for 15 min with 1% Triton X-100 (Sigma, Cat. #T9284) in PBS, blocking for 30 min with

5% skim milk (Megmilk Snow Brand) in TBS-T (50 mM tris(hydroxymethyl) aminomethane (Tris)-HCl pH 8.0, 150 mM NaCl and 0.05% Tween 20) and incubation at 4 °C overnight with the primary antibodies in antibody-dilution buffer (TBS-T supplemented with 5% bovine serum albumin (BSA; Iwai Chemicals, Cat. #A001) and 0.1% NaN$_3$ (Nacalai Tesque, Cat. #312-33)). The cells were further incubated at room temperature in the dark for 1–2 h with the appropriate fluorophore-conjugated secondary antibodies in TBS-T. After counterstaining with Hoechst 33258 dye (Dojindo, Cat. #343-07961, 1:2,000) in TBS-T for 5 min, the cover slips were mounted on glass slides with Fluoromount/Plus (Diagnostic Biosystems, Cat. #K048). The samples were observed by using an LSM 510 META (Zeiss) or a TCS SP5 microscope with the 63×/1.40 oil objective. To distinguish the background fluorescence or the "bleed-through" of the other fluorophore from the true signal, we also confirmed the proper negative control samples in each observation.

**Immunoelectron microscopy using ultrathin cryosections**. After Venus-ASK3-HEK293A cells were exposed to hyperosmotic stress (800 mOsm, 3 h), the cells were fixed at room temperature for 10 min with 4% paraformaldehyde (PFA) in 0.1 M phosphate buffer (pH 7.2) (PB), followed by the replacement with fresh 4% PFA in PB and incubation at 4 °C overnight. Cells were washed three times with PBS, followed by 0.15% glycine in PBS, and embedded in 12% gelatin in 0.1 M PB. Small blocks were rotated in 2.3 M sucrose in PB at 4 °C overnight and quickly plunged into liquid nitrogen. Sections ~60-nm thick were cut using a UC7/FC7 ultramicrotome (Leica) and picked up with a 1:1 mixture of 2% methylcellulose and 2.3 M sucrose in PB. The sections were incubated at 4 °C overnight with rabbit anti-GFP antibody (Frontier Institute, Ishikari, Japan, Cat. #GFP-Rb-Af2020), followed by incubation at room temperature for 1 h with protein A conjugated to 10-nm gold particles (protein A-gold; Cell Microscopy Center, University Medical Center Utrecht, Utrecht, the Netherlands). The sections were embedded in a thin layer of 2% methylcellulose with 0.4% uranyl acetate (pH 4.0) and observed with a H-7100 (Hitachi) transmission electron microscope. For control experiments, ultrathin sections were reacted only with protein A-gold.

**In vitro condensation assay**. The purified EGFP-FLAG-tagged protein was diluted into a sample in a microtube, whose control conditions were 10 µM EGFP-FLAG-tagged protein, 150 mM NaCl, 20 mM Tris (pH 7.5), and 1 mM dithiothreitol (DTT). When increasing macromolecular crowding, Ficoll PM400 (GE Healthcare, Cat. #17-0310-10) or PEG 4000 (Kanto Kagaku, Cat. #32828-02) was included in a sample at the indicated concentration. When modifying ion strength and concentration, the concentration of NaCl was changed as indicated. When investigating the effects of PAR or RNA on ASK3 condensates, 20% PEG was added as a crowding reagent, and the indicated concentration of PAR polymer (Trevigen, Cat. #4336-100-01), β-NAD (Sigma-Aldrich, Cat. #N7004), poly(A) RNA (Roche, Cat. #10108626001) or adenosine 5′-monophosphate (AMP; Sigma-Aldrich, Cat. #A1752) in TE (10 mM Tris-HCl pH 8.0 and 1 mM ethylenediaminetetraacetic acid (EDTA)) was also added. The prepared sample was subsequently incubated at 4 °C for 15 min. The reaction mixture was immediately loaded into a counting chamber with a cover slip (Matsunami, Cat. # C018241), followed by observation using a TCS SP5 microscope with a 63×/1.40 oil objective. To maintain a constant focal plane even if there were no condensates, we set the focal plane adjacent to the surface of the cover slip by utilizing the motorized stage and the on-demand mode of adaptive focus control system. Images of the EGFP signal were captured from five random fields per sample. Of note, we began from the protein purification procedures in each independent experiment.

For presentation, representative raw images were adjusted in brightness and contrast linearly and equally within the samples using Fiji/ImageJ software. For quantification, we established a macro script in Fiji/ImageJ to calculate the fluorescent intensity and area of ASK3 condensates in each sample and applied the script to all raw image sets in batch mode. In the script, a Gaussian filter and background correction were applied to each image, followed by particle analysis. Each parameter was determined from pilot analyses for EGFP-FLAG-ASK3 WT. The exported data table was further summarized in RStudio software. In the R script, the amount of ASK3 condensates in a sample was defined as the mean of total intensity within five fields. When investigating the effects of PAR, the amount value was divided by the amount of the internal standard sample, i.e., the control sample without TE addition. When comparing the effects of PAR between ASK3 mutants, the normalized amount value was further converted to the relative to the mean of control samples between experiments.

**Immunoblotting**. Cells were lysed in lysis buffer (20 mM Tris-HCl pH 7.5, 150 mM NaCl, 10 mM EDTA, 1% sodium deoxycholate, and 1% Triton X-100) supplemented with protease inhibitors (1 mM phenylmethylsulfonyl fluoride (PMSF) and 5 µg mL$^{-1}$ leupeptin). When detecting the phosphorylation of endogenous proteins, phosphatase inhibitor cocktail II (20 mM NaF, 30 mM β-glycerophosphatase, 2.5 mM Na$_3$VO$_4$, 3 mM Na$_2$MoO$_4$, 12.5 µM cantharidin and 5 mM imidazole) was also supplemented. When detecting the PARsylated proteins, 1 mM nicotinamide (Sigma-Aldrich, Cat. #N0078) and 100 µM gallotanin (Sigma-Aldrich, Cat. #403040) were also supplemented as PARP and PARG inhibitors, respectively. Cell extracts were clarified by centrifugation at 4 °C and ~16,500 × g for 15 min, and the

supernatants were sampled by adding 2× sample buffer (80 mM Tris-HCl pH 8.8, 80 µg mL$^{-1}$ bromophenol blue, 28.8% glycerol, 4% sodium dodecyl sulfate (SDS) and 10 mM DTT). After boiling at 98 °C for 3 min, the samples were resolved by SDS-PAGE and electroblotted onto a BioTrace PVDF (Pall), FluoroTrans W (Pall) or Immobilon-P (Millipore, Cat. #IPVH00010) membrane. The membranes were blocked with 2.5% or 5% skim milk in TBS-T and probed with the appropriate primary antibodies diluted by the antibody-dilution buffer (each dilution rate is summarized in Supplementary Data 1). After replacing and probing the appropriate secondary antibodies diluted with skim milk in TBS-T, antibody-antigen complexes were detected on X-ray films (FUJIFILM, Cat. #47410-07523, #47410-26615 and #47410-07595) using an enhanced chemiluminescence system (GE Healthcare). The films were converted to digital images by using a conventional scanner without any adjustment.

For presentation, representative images were acquired by linearly adjusting the brightness and contrast using GIMP software. When digitally trimming superfluous lanes from blot images, the trimming procedure was executed after the adjustment of brightness and contrast, and the trimmed position was clearly indicated. Quantification was performed against the raw digital images with densitometry using Fiji/ImageJ software. "Kinase activity", "Interaction" and "PARsylated proteins" were defined as the band intensity ratio of phosphorylated protein to total protein, the band intensity ratio of coimmunoprecipitated protein to input protein and the band intensity ratio of PARsylated proteins to actin, respectively. For the kinase activity of endogenous ASK3, the ratio of phosphorylated ASK to ASK3 was calculated.

**Coimmunoprecipitation assay**. The supernatants of cell extracts were incubated with anti-FLAG antibody beads (Wako Pure Chemicals Industries, clone 1E6, Cat. #016-22784) for 30–120 min at 4 °C. The beads were washed twice with wash buffer (20 mM Tris-HCl pH 7.5, 500 mM NaCl, 1% sodium deoxycholate, 1% Triton X-100, 0.2% SDS) and once with lysis buffer, followed by the addition of sample buffer.

**Computational simulation**. To simulate with the computational model (see Supplementary Note) in silico, we computed random trajectories of both ASK3 units and obstacles using the rejection kinetic Monte Carlo method in Python language. Briefly, the Phyton script executed the following algorithm. First, all ASK3 units and obstacles were randomly located in the grid space as an initial condition. Next, a target molecule with a chance to move was randomly picked from the union of the ASK3 unit and obstacle populations, followed by the random selection of a potential destination position of the selected target. As described in Supplementary Note, the designation of the target and destination systematically assigned $k$, the rate constant of the potential movement. At the same time, a random number $r$ was acquired from the interval [0, 1). If $k > r$, the movement was accepted, and the target was renewed at the position of destination, although the target was "renewed" at the original position in cases of exchange/vibration and reflection. Otherwise, if $k ≤ r$, the movement was denied, and the target stayed at the original position. After this determination, the iteration step number was incremented by one, and the next iteration step began by randomly selecting the next target molecule.

In the simulation, we regarded $k_4$, $k_6$ and $k_7$ as dummy constants, and we ignored their calculations for the reflection processes because the target molecule remained at its original position in all cases. Since the goal of the computational simulation in this study was not to understand the phase of condensates but to understand the driving force of ASK3 condensation, we also regarded $k_2$ as a dummy constant and ignored its calculation for the exchange/vibration process. We set $k_1 = 1$; therefore, free diffusion of ASK3 was always accepted. In contrast, we set $k_5 = 0.01$; therefore, the diffusion of an obstacle was slower than diffusion of ASK3 unit. This assumption is not unusual because each virtually integrated obstacle is considered the average movement of the constituent molecules, whose movement vectors mostly cancel each other out. For the penalty-defining constant $k_3$, we set $A = 1$ to adjust $k_3 = 1$ under the condition when $n_{lost} = 0$ and fixed $ΔE/θ = 1$ for simplicity; hence, the maximum speed of ASK3 for moving within the condensate was identical to $k_1$ (=1), the free diffusion of ASK3 out of condensate. Of note, the sampling distribution of $k_3$ in our simulation satisfied exponential decay curve, that is, rejection sampling was confirmed. We prepared 500 ASK3 units with or without 1500 obstacles in the grid space. To create the condition under osmotic stress, we changed only the grid space, ranging from 50 × 50 to 120 × 120 squares.

For figure presentation, our Python script saved the coordinates of each molecules, which was rendered using RStudio software. For movie presentation, our Python script saved representative images at every 10,000 or 100,000 iteration steps, defined as the unit of time $t$. To make a time-lapse video, a series of images with added captions were converted to a movie file using Fiji/ImageJ software. For quantification, our Python script calculated the count and size of the ASK3 clusters at every iteration step. To mimic confocal microscopy observations, a cluster of condensates was defined as a cluster of ≥6 consecutive ASK3 units. The exported data table was summarized in RStudio software.

**Data analysis and statistical analysis**. The data are summarized as the mean ± SEM with the exception of boxplots (center line: median, box limits: $Q_1$ and $Q_3$, whiskers: max(minimum value, $Q_1 − 1.5 × IQR$) and min(maximum value,

$Q_3 + 1.5 \times IQR$)). No statistical method was utilized to predetermine the sample size because all experiments in this study were performed with defined laboratory reagents and cell lines. Based on several pilot experiments to determine the experimental conditions, each sample size was chosen as large as possible to represent experimental variation while still practically feasible in terms of data collection. Based on the small sample size and the quality-oriented immunoblotting assays, homoscedasticity was assumed unless $P < 0.01$ in an $F$-test or Levene's test based on the absolute deviations from the median. For null hypothesis testing, statistical tests, the number of samples, the sample sizes and other statistic values are indicated in Supplementary Data 2. The statistical tests were performed using R with RStudio software, and $P < 0.05$ was considered statistically significant. In several experiments, a few samples were excluded because they satisfied the criteria clearly outlined in the above sections. For sample randomization, the independent experiments were performed across different passages of cells, and the cells in the control and treated groups were seeded from the same population of cells. The investigators were not blinded to allocation during experiments and outcome assessments. Nevertheless, the assessments were semi-automated with macro scripts in Fiji/ImageJ and R scripts, and there was little room for subjective bias.

**Reporting summary**. Further information on research design is available in the Nature Research Reporting Summary linked to this article.

## Data availability
The authors declare that all data supporting the findings of this study are available within the paper and its Supplementary Information files. Further information and requests for resources and reagents should be directed to K.W. and H.I. Source Data are provided with this paper.

## Code availability
Python scripts for the computational simulations are provided in Supplementary Software 1. All other custom scripts used in this study are fundamental enough for standard users to code the procedures described in the "Methods" section, but further information and requests for the custom scripts should be directed to K.W. and H.I.

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

## Acknowledgements

We thank H. Seimiya (Japanese Foundation for Cancer Research) for kindly providing TNKS cDNA; M. Davidson (Florida State University) for providing tdTomato cDNA via Addgene; all current and former members of the Laboratory of Cell Signaling for valuable materials and fruitful discussions. This work was supported by the Japan Agency for Medical Research and Development (AMED) under the Project for Elucidating and Controlling Mechanisms of Aging and Longevity (grant number JP20gm5010001 to H.I.), by the Japan Society for the Promotion of Science (JSPS) under the Grants-in-Aid for Scientific Research (KAKENHI; grant numbers JP18H03995 to H.I., JP18H02569 to I.N. and JP19K16067 to K.W.) and by the Japan Science and Technology Agency (JST) Moonshot R&D—MILLENNIA Program (grant number JPMJMS2022-18 to H.I.).

## Author contributions

K.W., I.N. and H.I. conceptualized and supervised this project. K.W. designed and performed almost all experiments. K.W. and K.M. established the computational model, and K.M. performed the computational simulation. K.M., X.Z. and S.S. helped with experiments. Y.U. and M.K. performed the TEM analysis. K.W. and K.M. performed the experiments for revision. K.W. statistically analyzed and visualized all data and illustrated all schematic figures. K.W., I.N. and H.I. wrote and revised the manuscript.

## Competing interests

The authors declare no competing interests.
