## [Peer Review File · Nature Communications]

REVIEWER COMMENTS

Reviewer #1 (Remarks to the Author):

Cells recognize osmotic stress through liquid-liquid phase separation lubricated with poly(ADP-ribose)

Watanabe and colleagues report their studies on the cellular response to osmotic stress. They focus on the ASK3 kinase as a reversible sensor of osmotic stress. The phosphorylation/de-phosphorylation of ASK3 correlates with the formation/disappearance of ASK3 puncta in cells. In a search for other factors that regulate this process, an siRNA screen identified NAMPT as a modulator. As NAMPT regulates cellular NAD⁺ levels, its overexpression or knockdown can regulate NAD⁺ levels, and in turn the activities of NAD⁺ utilizing enzymes, such as the PARP enzymes that produce poly(ADP-ribose). The study modulates NAD⁺ levels to alter poly(ADP-ribose) levels in cells, or modulates poly(ADP-ribose) levels directly by overexpressing PARG, an enzyme that degrades poly(ADP-ribose). These alterations affect the phosphorylation state of ASK3, suggesting a connection between ASK3 activation and NAD⁺/poly(ADP-ribose) levels. In vitro, poly(ADP-ribose) was able to disrupt EGFP-ASK3 condensates, suggesting the model that poly(ADP-ribose) alters the dynamics of ASK3 condensates.

The presented experiments are nicely done and the study addresses an interesting area. However, the connection between ASK3 puncta in cells and poly(ADP-ribose) production/binding is not really developed, so the study seems a little early stage for publication. There is no final model presented for the study, and this probably reflects that there is not a lot of mechanism developed for the association between NAD⁺/poly(ADP-ribose) levels and ASK3 condensate formation/activation. It still seems that some key questions should be addressed to develop this model. Which PARP is making the poly(ADP-ribose)? There are only a few PARP family members that make poly(ADP-ribose), as opposed to mono-ADP-ribose. The cytoplasmic location of the ASK3 puncta should also limit the likely PARPs. What is the signal for poly(ADP-ribose) production? Regarding the in vitro disruption of EGFP-ASK3 condensates, does RNA or DNA have the same affect as poly(ADP-ribose)?

Reviewer #2 (Remarks to the Author):

Watanabe et al study a potential link between physicochemical changes in the cell and signal transduction. Specifically, they study how osmotic changes affect the protein kinase ASK3/MAP3K5. They had previously reported that this kinase is phosphorylated under hypo-osmotic conditions and becomes dephosphorylated under hyperosmotic stress. Here they report that EGFP-ASK3 forms granules under hyperosmotic stress and number and size are dependent on the degree of stress. Employing a mathematical model and experimental data they provide as explanation that macromolecular crowding causes the formation of those granules, indicating a possible mechanism for intracellular sensing of osmotic stress. Performing mutational analysis of ASK3 they correlate ability for formation of condensates with the ability for dephosphorylation of ASK3, which then could affect downstream signalling. Employing results from a previously performed siRNA screen

they identify the NAD salvage pathway as regulator of ASK3 phosphorylation. They move on to show that this effect is mediated via PARsylation (polyADP-ribosylation), which previously had been shown to mediate protein condensation. However, in this case it rather appears that PARsylation somehow keeps PAK3 condensation in a state (“lubricates”) that makes the protein accessible for dephosphorylation by PP6.

This is an extremely interesting paper for the reason mentioned above: it provides evidence that links physicochemical changes in the cell to signalling and provides a possible osmostress sensing mechanism. The manuscript is rich in remarkably interesting data and is extremely well documented. Given the fact that the manuscript addresses fundamental biology, its richness in data and the tantalising hypothesis that it phrases it probably should have been considered in a journal of even higher impact.

This said, there are some instances where it appears that statements are not fully supported by the data presented, as further outlined below in the specific comments. It also appears that the mechanism proposed here should be discussed in more depth.

1. The authors focus in this study for apparent reasons on ASK3. Is the observed condensation of ASK3 a unique feature of this protein or do other related protein kinases also show such effects? The answer to this question would obviously provide evidence for how the cells translates a general signal (macromolecular crowding) into a specific response (mediated by ASK3).

2. The authors state: “Although exhibiting lower basal activities under isoosmotic conditions, ASK3 ΔC and $\Delta CCC\Delta CLCR$ were not inactivated under hyperosmotic stress (Fig. 2D and S3A), suggesting that ASK3 condensation is required for its inactivation.” There is a correlation between inability to condense and to become dephosphorylated. While this implies that condensation is a prerequisite for dephosphorylation it does not demonstrate this beyond doubt. The authors should rephrase their statements in this regard throughout the manuscript.

3. Overall, the hypothesis is that hyperosmotic causes macromolecular crowding, which causes ASK3 condensation, which provides a signal for ASK3 dephosphorylation by PP6. However, in order for PP6 to access ASK3 it is required to be “lubricated” by PARsylation. As mentioned above, this is an interesting, attractive hypothesis but also quite a complicated mechanism. Why would it be that complicated? Would the region of ASK3 required for condensation be a bona fide osmosensor? How is PARsylation regulated under osmostress in a crowded cell? And PP6 activity? The authors should elaborate in their discussion on those (and possibly other) question concerning the underlying mechanisms.

4. The authors throughout the paper write about effects on ASK3 activity when they actually only measure ASK3 phosphorylation. While changes in the phosphorylation state are likely to affect activity, this is not what they measure. Hence the statements on activity should be changed to phosphorylation.

5. In extension to that, the author do not state in the manuscript which antibody was used to detect phospho-ASK3. Probably it is an antibody specific to the form of ASK3 phosphorylated in its kinase

domain.

6. Fig. 2D: There is a statement on an unspecific band in the legend which is not indicated in the figure.

7. Fig. 2D: It appears that cells expressing the ASK3 deltaC construct do not express the wild type protein. Why is that?

8. The term “suppress” in the context of inhibition or reduction (of for instance the level of phosphorylation) may be irritating, because in genetics suppression has a specific meaning (suppressor mutations). While it is appreciated that the term may be used differently in different scientific communities the alternative terms above may be more suitable.

9. It appears as if the manuscript has been prepared for a different journal with restrictions on text size and number of figures. This causes a perhaps somewhat unnatural distribution of figures in the main text and the supplement and keeping introduction and discussion unnecessarily short. The authors should consider re-organising the manuscript to better match a typical paper in Nature Communications and to make the manuscript more accessible to general readers.

Reviewer #3 (Remarks to the Author):

Watanabe and colleagues address the fundamental, yet poorly understood question of how cells sense osmotic stress – using the tonicity-regulated kinase ASK3/MAP3K15 as a model system. Specifically, the authors follow-up on their previous observation that ASK3 is active only under hypotonic conditions (where it acts as a negative regulator of the WNK1-SPAK1 signaling axis), but inactive under hypertonic conditions. Intriguingly, they find that ASK3 undergoes liquid-liquid phase separation (LLPS) under hypertonic conditions and propose a model according to which LLPS inactivates ASK3 by trapping it into biomolecular condensates where it can be dephosphorylated by PP6. This inactive, phase-separated state of ASK3 can further be modulated by poly(ADP)-ribosylation (PARylation), a post-translational modification that had been linked to condensates before. The authors hint (but do not convincingly show) that PP6 can only be recruited to liquid-like, PARylated ASK3 condensates to fully inactivate the kinase.

Despite this shortcoming, I am overall excited by the work of Watanabe et al., because (a) it connects innate biophysical responses of cells to environmental perturbations with specific cellular adaptation mechanisms and (b) underscores that osmotic stress is emerging as a new paradigm to understand the multi-faceted ways that cells use phase separation to adapt to detrimental conditions, including activation and inhibition of cellular signaling processes (e.g. Yap1 and the present work of Watanabe et al., respectively). Thus, I think that the manuscript is ideally suited for the broad readership of Nature Communications.

Major points:

1.) The authors show that PARylation ‘lubricates’ ASK3 condensates and that lack of this lubrication

by PAR depletion or PAR-binding motif (PBM) mutants causes ASK3 condensates to solidify. However, the authors do not convincingly integrate these findings in their model that ASK3 condensation is required for its inactivation under hypertonic stress. In the discussion, the authors state that “PAR provides the opportunity for interaction between ASK3 and PP6 followed by ASK3 inactivation” (page 7, lines 269-270), but do not really show this even though this would be a key experiment to make their study truly go full circle. They indirectly touch upon this with the experiments in Fig.3G/H, but the way this experiment is introduced confused me rather than convinced me about their model. For example, it would be really nice if the authors could use their ASK3 PBM mutants to look at PP6 interaction in their biochemical and imaging assays. Even though I am aware that the Covid-19 pandemic poses restrictions on wet lab work, I think an experiment like this could beautifully tie together the previous work of the authors on ASK3 regulation and their LLPS model presented here. To be clear: I really like this work and primarily hope that my feedback helps to make it more convincing and more digestible.

2.) It would be nice if the authors could experimentally show that crowding induced independently of osmotic perturbations interferes with ASK3 activity (e.g. by interfering with the mTOR pathway: Delarue et al 2018 Cell).

Minor points:

- Page 4, lines 148-157: As mentioned above, I'm intrigued but confused by the PP6 connection. It seems that authors somehow want to distinguish between multiple levels of ASK3 inactivation. What I find striking is that oftentimes, multimerization of kinases leads to activation via auto-phosphorylation. However, here it seems that condensation causes recruitment of a phosphatase that presumably leads to dephosphorylation, which I find conceptually very interesting. I think the authors should revisit this paragraph and explain the logic behind their experiments better. I appreciate the concise manuscript, but I feel here clarity suffered a bit. Especially because there might also be a link to phase properties here, e.g. do ASK3 condensates have different material properties in PP6 knock-out/knock-down cells?

- Page 5, lines 178-199: Two things are really confusing here. First, the brief hint at PP6 before shifting focus entirely to PAR. Second, highlighting the contrast between NAMPT (NAD supplier) and CD38/SIRT2/PARP1 (NAD user) overexpression, because it seems to me that the effects are as expected based on the antagonistic roles of the proteins in the NAD cycle. The PP6 angle has been explored by the authors before, so perhaps it would be better at this point to simply conceptually point out here that NAD could indirectly effect ASK3 condensation via changes in redox state (which the authors do not explore) or more directly via NAD-dependent enzymatic reactions – which the authors explore, especially because there is already precedence for an effect of PARylation on condensates in the literature. As mentioned above, introducing the PP6 angle would make much more sense after the lubrication effect has been described.

Figures:

- Fig.1: There are a lot of different experimental approaches and really nice data cramped in this figure. It would help if this would be split up into multiple figures so that individual panels could be larger.
- Fig.1C/G: It would be easier to see the ASK3 clusters if they were blue instead of red.
- Fig. S5: There is a lot of nice data in this figure and it is prominently referenced in the main text. Perhaps at least parts of this supplement should be moved to a main figure?
- I think it would help to include a model figure at the end.

Point-by-point response

Dear all reviewers,

We appreciate your kind and thoughtful comments to our manuscript. We have to apologize for our lack of explanations in the initial manuscript, especially in the details about model mechanisms. In this revised manuscript, we leveraged the style of *Nature Communications* and mainly improved discussions and figure presentations. All texts accompanied with this reorganization were colored by blue, and the texts highly related to the reviewers' comments were also highlighted by yellow marker. Although we fully agree that there are somewhat unclear parts in our study, such as PAR regulation, we consider that these parts themselves can be future interesting topics; and we also described these potentials in Discussion section to be provided for the broad scientific communities.

Of note, during the reviewing period, the other group reported a similar observation with one of our findings: the molecular crowding-induced biomolecular condensates under hyperosmotic stress (Jalihal, A. P. et al. *Mol. Cell* 2020). Unlike their experimental model about the P-body-independent DCP1A, we focused on a functional osmosensitive kinase which is important for the rapid cellular response against osmotic stress. Therefore, our study combined with a computational model still have strong novelties; rather, their report would support our findings.

We hope that your concerns are relieved in this revised manuscript.

Sincerely yours,
Kengo Watanabe and Hidenori Ichijo

REVIEWER COMMENTS

Reviewer #1 (Remarks to the Author):

Cells recognize osmotic stress through liquid-liquid phase separation lubricated with poly(ADP-ribose)

Watanabe and colleagues report their studies on the cellular response to osmotic stress. They focus on the ASK3 kinase as a reversible sensor of osmotic stress. The phosphorylation/de-phosphorylation of ASK3 correlates with the formation/disappearance of ASK3 puncta in cells. In a search for other factors that regulate this process, an siRNA screen identified NAMPT as a modulator. As NAMPT regulates cellular NAD⁺ levels, its overexpression or knockdown can regulate NAD⁺ levels, and in turn the activities of NAD⁺ utilizing enzymes, such as the PARP enzymes that produce poly(ADP-ribose). The study modulates NAD⁺ levels to alter poly(ADP-ribose) levels in cells, or modulates poly(ADP-ribose) levels directly by overexpressing PARG, an enzyme that degrades poly(ADP-ribose). These alterations affect the phosphorylation state of ASK3, suggesting a connection between ASK3 activation and NAD⁺/poly(ADP-ribose) levels. In vitro, poly(ADP-ribose) was able to disrupt EGFP-ASK3 condensates, suggesting the model that poly(ADP-ribose) alters the dynamics of ASK3 condensates.

The presented experiments are nicely done and the study addresses an interesting area. However, the connection between ASK3 puncta in cells and poly(ADP-ribose) production/binding is not really developed, so the study seems a little early stage for publication. There is no final model presented for the study, and this probably

reflects that there is not a lot of mechanism developed for the association between NAD⁺/poly(ADP-ribose) levels and ASK3 condensate formation/activation. It still seems that some key questions should be addressed to develop this model. Which PARP is making the poly(ADP-ribose)? There are only a few PARP family members that make poly(ADP-ribose), as opposed to mono-ADP-ribose. The cytoplasmic location of the ASK3 puncta should also limit the likely PARPs.

Response:

We completely agree with the importance of PARP(s) identification to advance our study. As the reviewer suggested, PARP1, PARP2, TNKS and TNKS2 are currently said to have the ability of PARsylation (Hottiger, M. O. *Mol. Cell* 2015; Barkauskaite, E. et al. *Mol. Cell* 2015; Leung, A. K. L. *Trends Cell Biol.* 2020), and we have investigated whether these PARPs regulates ASK3 dephosphorylation/inactivation under hyperosmotic stress. However, (1) the overexpression of TNKS or TNKS2 inhibited ASK3 dephosphorylation under hyperosmotic stress in a PARP activity-dependent manner (Supplementary Fig. 6e, f); (2) the PARP1 overexpression inhibited ASK3 dephosphorylation under hyperosmotic stress (Supplementary Fig. 6d); (3) a PARP inhibitor 1,5-dihydroxyisoquinoline (DiQ) whose broad spectrum covers PARP1 and PARP2 (Wahlberg, E. et al. *Nat. Biotechnol.* 2012) enhanced ASK3 dephosphorylation under hyperosmotic stress (Supplementary Fig. 6g); (4) our genome-wide siRNA screen identified no known PARP family member as an ASK3 inactivator candidate. Therefore, we have not succeeded to identify the responsible PARP(s) in ASK3 inactivation thus far. Nevertheless, in addition to our results using NAMPT or PARG for the alternative interventions of PAR dynamics (Fig. 5), we have noticed that a PAR-reading E3 ubiquitin ligase RNF146 was included in our genome-wide siRNA screen (Fig. 4a) and already confirmed that PAR-recognition ability of RNF146 is required for ASK3 inactivation (our unpublished data), suggesting that PAR has an important role in ASK3 regulation at least. As one of the reasons why we cannot have identified the responsible PARP(s), we speculate that the responsible PARPs function redundantly and cooperatively in ASK3 regulation. Or, when considering that a novel NADase enzyme has recently been identified even in this modern biology (Essuman, K. et al. *Neuron* 2017), there may be still unidentified PARPs. Because the PAR regulation itself is an interesting topic in perspective of the PAR biology, we believe that this part should be approached in the future works. In this revised manuscript, we added the sentences and figures related to this response (lines 241–243 of page 6, lines 410–422 of page 10, Supplementary Fig. 6e–g). We also included our final model in this study (Fig. 7) and discussions about it (lines 355–383 of page 9).

What is the signal for poly(ADP-ribose) production?

Response:

We appreciate the reviewer's thoughtful question. Through our analyses, we noticed that hyperosmotic stress slightly increases global PAR level (Point-by-point response Fig. 1). However, it is well known that hyperosmotic stress induces oxidative stress (Burg, M. B. et al. *Physiol. Rev.* 2007) and that oxidative stress results in PARP1 activation (Bai, P. *Mol. Cell* 2015). Therefore, we cannot conclude whether the responsible PAR in ASK3 inactivation is increased until the responsible PARP is identified. At the same time, although PAR can be synthesized and degraded within seconds, we are leaning toward the possibility that the basal level of PAR is sufficient for the liquidity of ASK3 condensates. We added the sentences related to this response (lines 422–428 of page 10).

Regarding the in vitro disruption of EGFP-ASK3 condensates, does RNA or DNA have the same affect as poly(ADP-ribose)?

Response:

We appreciate the reviewer's ingenious question. We investigated the effects of RNA on ASK3 condensates in vitro. The results showed that poly(A) RNA also inhibits the formation of solid-like ASK3 condensates in vitro (Point-by-point response Fig. 2), suggesting that the physicochemical property of PAR is important for the liquidity maintenance of ASK3 condensates. Nevertheless, please note that the chain length of RNA we used is quite longer than that of PAR according to the manufactures' datasheets, which would be more advantage for RNA to interact with ASK3s multivalently and to melt the solid-like ASK3 condensates. We do not exclude the possibility that RNA also regulates ASK3 condensates under hyperosmotic stress, but we have not had any data judging whether RNA regulates ASK3 condensates in cells. Hence, we would like to focus on PAR in this study.

Reviewer #2 (Remarks to the Author):

Watanabe et al study a potential link between physicochemical changes in the cell and signal transduction. Specifically, they study how osmotic changes affect the protein kinase ASK3/MAP3K5. They had previously reported that this kinase is phosphorylated under hypo-osmotic conditions and becomes dephosphorylated under hyperosmotic stress. Here they report that EGFP-ASK3 forms granules under hyperosmotic stress and number and size are dependent on the degree of stress. Employing a mathematical model and experimental data they provide as explanation that macromolecular crowding causes the formation of those granules, indicating a possible mechanism for intracellular sensing of osmotic stress. Performing mutational analysis of ASK3 they correlate ability for formation of condensates with the ability for dephosphorylation of ASK3, which then could affect downstream signalling. Employing results from a previously performed siRNA screen they identify the NAD salvage pathway as regulator of ASK3 phosphorylation. They move on to show that this effect is mediated via PARsylation (polyADP-ribosylation), which previously had been shown to mediate protein condensation. However, in this case it rather appears that PARsylation somehow keeps PAK3 condensation in a state (“lubricates”) that makes the protein accessible for dephosphorylation by PP6.

This is an extremely interesting paper for the reason mentioned above: it provides evidence that links physicochemical changes in the cell to signalling and provides a possible osmotic stress sensing mechanism. The manuscript is rich in remarkably interesting data and is extremely well documented. Given the fact that the manuscript addresses fundamental biology, its richness in data and the tantalising hypothesis that it phrases it probably should have been considered in a journal of even higher impact.

This said, there are some instances where it appears that statements are not fully supported by the data presented, as further outlined below in the specific comments. It also appears that the mechanism proposed here should be discussed in more depth.

1. The authors focus in this study for apparent reasons on ASK3. Is the observed condensation of ASK3 a unique feature of this protein or do other related protein kinases also show such effects? The answer to this question would obviously provide evidence for how the cells translates a general signal (macromolecular

crowding) into a specific response (mediated by ASK3).

Response:

We fully understand the importance of the reviewer's question in point of generalization. In this study, we used ASK3 as an experimental model because we previously revealed that ASK3 is an important osmoresponsive kinase to orchestrate cell volume regulation in osmotic stress response. Now applying the paradigm of LLPS, we can reassess that another osmoresponsive kinase WNK1 also forms liquid-demixing condensates, which has been only regarded as WNK puncta or WNK bodies thus far although even FRAP of WNK puncta was reported (Zagórska, A. et al. *J. Cell Biol.* 2007; Sengupta, S. et al. *J. Biol. Chem.* 2012; Boyd-Shiwarski, C. R. et al. *Mol. Biol. Cell* 2018). In addition to the osmoresponsive kinases in cell volume regulation, the biomolecular condensates induced by LLPS just seconds after hyperosmotic stress, such as YAP condensates and proteasome foci, have been reported very recently (Cai, D. et al. *Nat. Cell Biol.* 2019; Yasuda, S. et al. *Nature* 2020). Furthermore, as mentioned in the opening sentences of this point-by-point response, the importance of molecular crowding in hyperosmotic stress-induced biomolecular condensates has been also proposed by another group very recently (Jalihal, A. P. et al. *Mol. Cell* 2020). Therefore, we think that our findings combined with our simple computational model can be extrapolated to the general cellular osmosensing mechanism. In this revised manuscript, we added these discussions (lines 301–306 of pages 7–8).

2. The authors state: “Although exhibiting lower basal activities under isoosmotic conditions, ASK3 ΔC and $\Delta CCC\Delta CLCR$ were not inactivated under hyperosmotic stress (Fig. 2D and S3A), suggesting that ASK3 condensation is required for its inactivation.” There is a correlation between inability to condense and to become dephosphorylated. While this implies that condensation is a prerequisite for dephosphorylation it does not demonstrate this beyond doubt. The authors should rephrase their statements in this regard throughout the manuscript.

Response:

We apologize for the insufficient logic. We completely agree with the reviewer's thoughtful comment only when reading our previous manuscript: Fig. 3b–d (corresponding to the previous Fig. 2B–D) indicate only the correlation between ASK3 condensation and ASK3 dephosphorylation. However, we found that (1) ASK3 condensation was normally observed under hyperosmotic stress even when ASK3 dephosphorylation was inhibited by the knockdown of PP6 (Supplementary Fig. 4a) and that (2) two kinase-inactive ASK3 mutants K681M and T808A did not form condensates under isoosmotic conditions (Supplementary Fig. 4b), suggesting that ASK3 inactivation is neither necessary nor sufficient for the condensate formation. Strictly speaking, we cannot exclude the possibility that there are confounding effects (e.g. ΔC and $\Delta CCC\Delta CLCR$ are just abnormally folded mutants). Nevertheless, we think that the combination of these data “suggests” that ASK3 condensation is required for ASK3 dephosphorylation. By utilizing the flexible format of *Nature Communications*, we added the sentences and figures related to this response (lines 180–187 of page 5, Supplementary Fig. 4).

3. Overall, the hypothesis is that hyperosmotic causes macromolecular crowding, which causes ASK3 condensation, which provides a signal for ASK3 dephosphorylation by PP6. However, in order for PP6 to access ASK3 it is required to be “lubricated” by PARsylation. As mentioned above, this is an interesting, attractive hypothesis but also quite a complicated mechanism. Why would it be that complicated? Would the region of ASK3 required for condensation be a bona fide osmosensor? How is PARsylation regulated under osmotic stress in a crowded cell? And PP6 activity? The authors should elaborate in their discussion on those (and possibly other) question concerning the underlying mechanisms.

Response:

We appreciate the reviewer's deep understanding and comments. In this revised manuscript, we

added the detail discussions about the molecular mechanism of ASK3 condensates in Discussion. Here, we particularly respond to the reviewer's specified questions as the followings.

As one of the reasons why cells have the "complicated" system, the liquid-to-solid transition of ASK3 condensates may also be a cellular functional response in a certain environment. We previously discovered that the insufficient inactivation of ASK3 under hyperosmotic stress leads to the failure of cell volume recovery, followed by the cell death (Watanabe, K. et al. *Cell Rep.* 2018). In a regulated necrotic cell death parthanatos, PAR is accumulated by PARP1 overactivation and trigger the translocation of apoptosis-inducing factor (AIF) (Fatokun, A. A. et al. *Br. J. Pharmacol.* 2014). Considering that PARP1 overexpression inhibited ASK3 dephosphorylation under hyperosmotic stress (Supplementary Fig. 6d), therefore, the regulation of PAR dynamics may efficiently link the phase regulation of ASK3 condensates with the cell death induction. This discussion was added in lines 401–409 of page 10.

As to the question about a bona fide osmosensor, we should first take into account the potentially multiple meaning of the term "osmosensor". In this study, we prefer to regard osmosensor at the cellular system level; that is, cells sense osmotic stress by macromolecular crowding-induced biomolecular condensates (as described in lines 309–311 and 316–320 of page 8). At the same time, we can also regard osmosensor at the molecular level; in this meaning, we would propose that ASK3 protein is one of osmosensors. Alternatively, it is also possible to regard osmosensor at the peptide motif/domain level, which would be the reviewer's intension. Unlike the proposed pH sensor of Sup35 (Franzmann, T. M. et al. *Science* 2018) or redox sensor of Pbp1 (Kato, M. et al. *Cell* 2019), we think that CCC and CLCR of ASK3 are just a constituent driving force for the condensate formation and prerequisite for osmosensing; rather, we think that the macromolecular crowding is the bona fide trigger of condensate formation under hyperosmotic stress. Furthermore, in case of ASK3, the PBM4 is important for the material property of condensates and the appropriate signal transduction, which also brings reluctance to regard osmosensor at the peptide motif/domain level. Therefore, we included the discussions about molecular mechanism from the point of driving force (lines 336–354 of pages 8–9).

As to the regulation of PARsylation, we have not succeeded to identify the responsible PARP(s) in ASK3 inactivation thus far, although we tried all known PARPs possessing PARsylation ability (Supplementary Fig. 6d–g; lines 410–422 of page 10). At least, hyperosmotic stress slightly increases global PAR level (Point-by-point response Fig. 1). However, it is well known that hyperosmotic stress induces oxidative stress (Burg, M. B. et al. *Physiol. Rev.* 2007) and that oxidative stress results in PARP1 activation (Bai, P. *Mol. Cell* 2015). Therefore, we cannot conclude whether the responsible PAR in ASK3 inactivation is increased until the responsible PARP is identified. At the same time, although PAR can be synthesized and degraded within seconds, we are leaning toward the possibility that the basal level of PAR is sufficient for the liquidity of ASK3 condensates. Because the PAR regulation itself is an interesting topic in perspective of the PAR biology, we believe that this part should be approached in the future works. At least, we added the discussions related to this response (lines 422–435 of page 10).

As to the phosphatase activity of PP6, we previously revealed that not the phosphatase activity of PP6 but the interaction between PP6 and ASK3 is increased under hyperosmotic stress, followed by the direct dephosphorylation of ASK3 by PP6 (Watanabe, K. et al. *Cell Rep.* 2018). While adding this sentence in lines 223–226 of page 6, we mainly discussed about the interaction between PP6 and ASK3 (lines 378–401 of pages 9–10).

4. The authors throughout the paper write about effects on ASK3 activity when they actually only measure ASK3 phosphorylation. While changes in the phosphorylation state are likely to affect activity, this is not what they measure. Hence the statements on activity should be changed to phosphorylation.

Response:

We appreciate the reviewer's advice. In the sentences describing results, we modified into the more accurate word "dephosphorylation", although we remained the word "inactivation" in some parts to make our intension clearer.

5. In extension to that, the author do not state in the manuscript which antibody was used to detect phospho-ASK3. Probably it is an antibody specific to the form of ASK3 phosphorylated in its kinase domain.

Response:

We appreciate the reviewer's comment. Although having stated in Methods, we added in the main text based on the importance of this information (lines 187–189 of page 5).

6. Fig. 2D: There is a statement on an unspecific band in the legend which is not indicated in the figure.

Response:

We improved the overlapping of the dagger sign with the arrowhead in Fig. 3d (corresponding to the previous Fig. 2D).

7. Fig. 2D: It appears that cells expressing the ASK3 deltaC construct do not express the wild type protein. Why is that?

Response:

In Fig. 3d (corresponding to the previous Fig. 2D), we used anti-FLAG antibody for checking the total amount of ASK3 constructs, which cannot detect endogenous wild-type ASK3. Because we added EGFP-FLAG-tag to each ASK3 construct, we can separately see the phosphorylation bands of endogenous ASK3 around 130 kDa in "IB: p-ASK (long exposure)", which is autophosphorylated from exogenously expressed ASK3s. We improved these presentations in Fig. 3d and 6c.

8. The term "suppress" in the context of inhibition or reduction (of for instance the level of phosphorylation) may be irritating, because in genetics suppression has a specific meaning (suppressor mutations). While it is appreciated that the term may be used differently in different scientific communities the alternative terms above may be more suitable.

Response:

We appreciate the reviewer's advice. Based on the broad readership of *Nature Communications*, we avoided to use "suppression" in this revised manuscript.

9. It appears as if the manuscript has been prepared for a different journal with restrictions on text size and number of figures. This causes a perhaps somewhat unnatural distribution of figures in the main text and the supplement and keeping introduction and discussion unnecessarily short. The authors should consider re-organising the manuscript to better match a typical paper in *Nature Communications* and to make the manuscript more accessible to general readers.

Response:

We are grateful to the reviewer's kind understanding. In this revised manuscript, we largely reorganized texts and figure presentations according to the style of *Nature Communications*.

Reviewer #3 (Remarks to the Author):

Watanabe and colleagues address the fundamental, yet poorly understood question of how cells sense osmotic stress – using the tonicity-regulated kinase ASK3/MAP3K15 as a model system. Specifically, the authors follow-up on their previous observation that ASK3 is active only under hypotonic conditions (where it acts as a negative regulator of the WNK1-SPAK1 signaling axis), but inactive under hypertonic conditions. Intriguingly, they find that ASK3 undergoes liquid-liquid phase separation (LLPS) under hypertonic conditions and propose a model according to which LLPS inactivates ASK3 by trapping it into biomolecular condensates

where it can be dephosphorylated by PP6. This inactive, phase-separated state of ASK3 can further be modulated by poly(ADP)-ribosylation (PARylation), a post-translational modification that had been linked to condensates before. The authors hint (but do not convincingly show) that PP6 can only be recruited to liquid-like, PARylated ASK3 condensates to fully inactivate the kinase. Despite this shortcoming, I am overall excited by the work of Watanabe et al., because (a) it connects innate biophysical responses of cells to environmental perturbations with specific cellular adaptation mechanisms and (b) underscores that osmotic stress is emerging as a new paradigm to understand the multi-faceted ways that cells use phase separation to adapt to detrimental conditions, including activation and inhibition of cellular signaling processes (e.g. Yap1 and the present work of Watanabe et al., respectively). Thus, I think that the manuscript is ideally suited for the broad readership of Nature Communications.

Major points:

1.) The authors show that PARylation ‘lubricates’ ASK3 condensates and that lack of this lubrication by PAR depletion or PAR-binding motif (PBM) mutants causes ASK3 condensates to solidify. However, the authors do not convincingly integrate these findings in their model that ASK3 condensation is required for its inactivation under hypertonic stress. In the discussion, the authors state that “PAR provides the opportunity for interaction between ASK3 and PP6 followed by ASK3 inactivation” (page 7, lines 269-270), but do not really show this even though this would be a key experiment to make their study truly go full circle. They indirectly touch upon this with the experiments in Fig.3G/H, but the way this experiment is introduced confused me rather than convinced me about their model. For example, it would be really nice if the authors could use their ASK3 PBM mutants to look at PP6 interaction in their biochemical and imaging assays. Even though I am aware that the Covid-19 pandemic poses restrictions on wet lab work, I think an experiment like this could beautifully tie together the previous work of the authors on ASK3 regulation and their LLPS model presented here. To be clear: I really like this work and primarily hope that my feedback helps to make it more convincing and more digestible.

Response:

We appreciate the reviewer’s deep understanding and thoughtful suggestion. We investigated the interaction between ASK3 PBM4 mutant and PP6 by using both immunoprecipitation (IP) and live-cell imaging. In IP experiments, we obtained the tendency that the hyperosmotic stress-dependent increase in the interaction between PP6 and ASK3 is diminished in the ASK3 PBM4 mutant (Point-by-point response Fig. 3). However, the increase in the PP6-ASK3 interaction was observed with the ASK3 PBM4 mutant in another trial; therefore, we cannot conclude the interaction ability of ASK3 PBM4 due to poor reproducibility. As one of possible explanations for this instability, we think that the PBM4 region is the most critical region for the interaction between PAR and ASK3, but the other regions also contribute for the multivalent interactions with PAR and/or the liquidity maintenance of ASK3 condensates. In fact, we can see that (1) ASK3 PBM4 mutant was not significantly but slightly

Point-by-point response Fig. 3 Hyperosmotic stress-dependent increase in the interaction between PP6 and ASK3 tended to be diminished in the ASK3 PBM4 mutant. Ability of the ASK3 PBM4 mutant to interact with ANKRD52 under hyperosmotic stress in HEK293A cells. ANKRD52: one of the PP6 subunit^{8,26}. The left panel is a “champion” image set of immunoblotting, and the right graph depicts the quantification of 2 independent experiments. Hyperosmolality (-): 300 mOsm; (+): 500 mOsm; 10 min. IB: immunoblotting, IP: immunoprecipitation. Data: mean.

inactivated under hyperosmotic stress (please see the quantified data; Supplementary Fig. 5j) and that (2) the FRAP of ASK3 PBM4 condensates were lower than that of WT but higher than that of CT (Fig. 6e and f). Furthermore, the IP experiment of PP6-ASK3 interaction is basically a “severe” experiment possibly due to the difficulty of holding the PP6-ASK3 interaction during in vitro operations. Please also note that PBM4 is a just “PAR-binding” motif and we have not reached to the conclusion what the target protein of PARylation is or even whether the free PAR is enough; hence, the noncovalent interaction between PAR and ASK3 may be fragile in vitro. We are afraid that our insufficient explanations led to the reviewer’s misunderstanding, and we also improved this point (Fig. 7b, lines 428–433 of page 10).

In live-cell imaging, the ASK3 PBM4 condensates was not separated from PP6 condensates; rather, the ASK3 PBM4 condensates well merged with PP6 condensates (Supplementary Fig. 7). This result may seem contrary to our conclusion that PAR maintains the liquidity of ASK3 condensates to make ASK3 inactivated by PP6. However, we currently hypothesize that ASK3 dephosphorylation occurs around the shared phase boundaries between PP6 and ASK3 condensates. Hence, this result that the ASK3 PBM4 condensates lost the shared phase boundaries between independent condensates accords with our model. Rationality of the strategy to use the surface vicinity of condensates in ASK3 dephosphorylation remains to be clarified, but this strategy may be advantageous according to the results that (1) the number and size of ASK3 condensates increases and decreases, respectively, under stronger hyperosmotic stress (Fig. 1b), indicating that the cellular total surface area of condensates increases in a hyperosmolality strength-dependent manner, and that (2) ASK3 dephosphorylation is enhanced in a hyperosmolality strength-dependent manner (Naguro, I et al. *Nat. Commun.* 2012). Since existence of the partly contacted condensates of PP6 and ASK3 suggests that the surface tensions from three interfaces, ASK3–cytosol, cytosol–PP6 and PP6–ASK3, are balanced (Shin, Y and Brangwynne, C. P. *Science* 2017), the PP6–ASK3 interface may be evolved to function as the private channel for PP6 and ASK3 within the limited space under hyperosmotic stress. In this revised manuscript, we added the result of live-cell imaging (Supplementary Fig. 7) and deep discussions of our model in point of PAR-dependent “lubrication” in ASK3 inactivation (lines 355–401 of pages 9–10).

2.) It would be nice if the authors could experimentally show that crowding induced independently of osmotic perturbations interferes with ASK3 activity (e.g. by interfering with the mTOR pathway: Delarue et al 2018 Cell).

Response:

We completely agree with the reviewer’s opinion. However, it is practically too difficult to operate only macromolecular crowding in cells. As the reviewer suggested, manipulations of the mTOR pathway may seem to be an option to regulate macromolecular crowding according to the previous paper (Delarue, M et al. *Cell* 2018); that is, it would be assumed that the activation of mTORC1 increases macromolecular crowding, followed by the condensate formation and inactivation of ASK3. However, the mTOR pathway has many important functions in cells. Especially, a mTOR inhibitor rapamycin activates ASK1 via its phosphatase inhibition (Huang, S. et al. *Mol. Cell* 2003; Huang, S. et al. *J. Biol. Chem.* 2004); therefore, manipulation of the mTOR pathway would affect ASK3 via ASK1. In addition, it is said that mTOR controls cell volume (Fingar, D. C. et al. *Genes Dev.* 2002). Therefore, we cannot simply regard the mTORC1 activation as the increase in macromolecular crowding, and we did not perform the suggested experiment. As mentioned in the opening sentences of this point-by-point response, the importance of molecular crowding in hyperosmotic stress-induced biomolecular condensates has been also proposed by another group very recently (Jalihal, A. P. et al. *Mol. Cell* 2020), and they also did not distinguish molecular crowding from other effects of osmotic perturbation. Nevertheless, because of the difficulty in cellular experiments, we developed the computational model to suggest the importance of macromolecular crowding. We would be grateful if the reviewer

could take this point into account.

Minor points:

- Page 4, lines 148-157: As mentioned above, I'm intrigued but confused by the PP6 connection. It seems that authors somehow want to distinguish between multiple levels of ASK3 inactivation. What I find striking is that oftentimes, multimerization of kinases leads to activation via auto-phosphorylation. However, here it seems that condensation causes recruitment of a phosphatase that presumably leads to dephosphorylation, which I find conceptually very interesting. I think the authors should revisit this paragraph and explain the logic behind their experiments better. I appreciate the concise manuscript, but I feel here clarity suffered a bit. Especially because there might also be a link to phase properties here, e.g. do ASK3 condensates have different material properties in PP6 knock-out/knock-down cells?

Response:

We apologize for the insufficient explanations. In this study, we described the role of ASK3 condensates in ASK3 dephosphorylation. However, we do not exclude the possibility that ASK3 condensates serve as multifunctional signaling hubs for the whole regulation of ASK3 activity more generally. This notion that ASK3 condensates provide condition-dependent signaling hubs may resolve the question why the autophosphorylation/activation of ASK3 is not predominant within the dense condensates under hyperosmotic stress. We improved the sentences about this point (lines 323–333 of page 8). As to the specified experiment, the knockdown of PP6 induces no apparent difference in ASK3 condensates under hyperosmotic stress (Supplementary Fig. 4a). And, we currently think that PP6 is not a modulator of ASK3 condensates but just an ASK3 phosphatase. Related to the above response to the major point 1, we added the model mechanism of the relationship between PP6 and ASK3 in lines 378–401 of pages 9–10, and we did not add the logic in the suggested paragraph based on the balance of whole manuscript.

- Page 5, lines 178-199: Two things are really confusing here. First, the brief hint at PP6 before shifting focus entirely to PAR. Second, highlighting the contrast between NAMPT (NAD supplier) and CD38/SIRT2/PARP1 (NAD user) overexpression, because it seems to me that the effects are as expected based on the antagonistic roles of the proteins in the NAD cycle. The PP6 angle has been explored by the authors before, so perhaps it would be better at this point to simply conceptually point out here that NAD could indirectly effect ASK3 condensation via changes in redox state (which the authors do not explore) or more directly via NAD-dependent enzymatic reactions – which the authors explore, especially because there is already precedence for an effect of PARylation on condensates in the literature. As mentioned above, introducing the PP6 angle would make much more sense after the lubrication effect has been described.

Response:

We apologize for our poor explanations. As a molecular mechanism of ASK3 dephosphorylation under hyperosmotic stress, we previously revealed that not the phosphatase activity of PP6 but the interaction between PP6 and ASK3 is increased under hyperosmotic stress, followed by the direct dephosphorylation of ASK3 by PP6 (Watanabe, K. et al. *Cell Rep.* 2018). Hence, we simply investigated whether NAD regulates ASK3 inactivation via PP6 interaction at this NAMPT subsection. And we mentioned about the results about NAD consumers after this subsection because the scientists in the NAD field would easily feel the logical gap without them. If we had succeeded to identify the responsible PARP(s) in ASK3 inactivation, we could connect NAD with PAR more smoothly. As the reviewer suggested, Fig. 4g and h could be moved after Fig. 5i, while the NAD experts may feel more uncomfortable about using NAMPT in the PAR subsection. Hence, we improved the sentences introducing Fig. 4g and h (lines 219–226 of page 6) and added the sentence to reminding readers of the results when discussing the model mechanism (lines 378–385 of page 9).

Figures:

- Fig.1: There are a lot of different experimental approaches and really nice data cramped in this figure. It would help if this would be split up into multiple figures so that individual panels could be larger.

Response:

Following the reviewer's advice, we separated this figure into Fig. 1 and 2 in this revised manuscript. In addition, we added the zoomed images and improved the figure representations in all figures.

- Fig.1C/G: It would be easier to see the ASK3 clusters if they were blue instead of red.

Response:

We tried making the blue-colored figure, but we did not feel it was easier to see. Instead, we improved the figure size (Fig. 1d and 2c).

- Fig. S5: There is a lot of nice data in this figure and it is prominently referenced in the main text. Perhaps at least parts of this supplement should be moved to a main figure?

Response:

Following the reviewer advice, we moved the figure into a main figure (Fig. 6).

- I think it would help to include a model figure at the end.

Response:

We added a model figure as Fig. 7.

REVIEWER COMMENTS

Reviewer #1 (Remarks to the Author):

Cells recognize osmotic stress through liquid-liquid phase separation lubricated with poly(ADP-ribose)

Watanabe and colleagues have largely addressed the concerns that I raised in the initial review, and they have added additional experiments to address the identity of the PARP responsible for producing poly(ADP-ribose). Although the responsible PARP still remains to be identified, I feel that they have added the appropriate comments to the text, and they have added to the Supplement the work that they have performed thus far.

Regarding point-by-point Fig. 2 that shows that RNA can inhibit ASK3 condensates in vitro, I feel that this data could be included in the study, but perhaps with a control experiment that shows that some length of RNA (or even just ribonucleotides) does not have the effect. And perhaps whether uncharged polymers, or positively charged polymers, have the same effect.

Reviewer #2 (Remarks to the Author):

The authors have fully and thoughtfully addressed my comments in the revised manuscript.
Stefan Hohmann

Reviewer #3 (Remarks to the Author):

Watanabe et al. carefully addressed the review comments during revision, providing new data in both the manuscript and the point-by-point reply. In my opinion, there are three key questions that came up during the revision:

1. Is the work still timely in light of recent publications?
2. Did the authors uncover the full mechanism?
3. Does the LLPS model of the authors for ASK3 regulation under hypertonic stress add up?

My take is:

1. Despite multiple recent reports of 'hyperosmotic phase separation (HOPS)', I think that the work of Watanabe et al. stands out because of their efforts to uncover the mechanism and functional meaning of (ASK3) HOPS, rather than merely describing the phenomenon. Thus, I still consider the work of Watanabe et al. timely, novel and exciting (perhaps it is also noteworthy in this regard that they shared their findings as a preprint approx. one month earlier than Jalihal et al.).

2. There has been great excitement about the manuscript during review, but also a consensus that the mechanisms has not yet been fully worked out. An obvious missing piece to the puzzle is the identity of the PARP that modifies ASK3 condensates. I do not think that the bar for publication

should be that the full regulatory pathway of a complex biological event, including all molecular players, has to be worked out in a single paper. The authors provide plenty of exciting and stimulating findings of interest to many fields (stress response biology, condensate function, kinase regulation, PARylation, ...) that will move science—a collective endeavor—forward.

3. This leaves the most important question: is the current working model of the authors plausible, i.e. backed by the data? To me, a key aspect of their model that hadn't been directly tested in the original manuscript is the whether PAR-dependent tuning of the ASK condensate material properties is truly required for PP6 recruitment to the condensate surface. In response to my comment, the authors performed an experiment with a perhaps unexpected, but nevertheless exciting result: PP6 is fully dispersed in PBM4 mutant condensates. This finding is very intriguing, but brings up a number of questions:

3a. Given that PBM4 mutant droplets are more solid, this suggests that the whole regulatory system may now be trapped in a dead-end aggregate (this could be further tested by two-channel FRAP). Such aggregates may work fine to inactivate ASK3 in response to a singular hypertonic insult even without complete dephosphorylation, but I wonder if the cells now lose their ability to respond to hypotonic stress – such as the cells experience when returning to isotonic from hypertonic?

3b. I also wonder what FK866 treatment does to wild type ASK3-PP6 condensates? I'm not sure that I agree that FK866 treatment truly "inhibited [...] the interaction between PP6 and ASK3 under hypertonic stress" (lines 226-228) based on the data shown in Fig.4g – it rather looks slightly reduced at best. Perhaps the phenotype on the condensate level is the same as for PBM4 and what is really going on is that lack of PAR causes ASK3 and PP6 to 'aggregate'?

3c. Thus, I still think that directly showing, on the condensate level by microscopy, that PARylation alters ASK3 and PP6 interaction would be required to make the condensate-based model of the authors fully convincing – especially as the authors themselves cast doubt on the IP approach when they speculate about "the difficulty of holding the PP6-ASK3 interaction during in vitro operations", which is a valid concern with condensate-based interactions.

Point-by-point response

Dear all reviewers,

We appreciate your time to review our revised manuscript and your kind comments. Based on the reviewers' comments, we re-revised our manuscript. All texts changed in this revision were colored by blue, and the texts highly related to the reviewers' comments were also highlighted by yellow marker. We hope that your concerns are relieved in this revised manuscript.

Sincerely yours,
Kengo Watanabe and Hidenori Ichijo

REVIEWER COMMENTS

Reviewer #1 (Remarks to the Author):

Cells recognize osmotic stress through liquid-liquid phase separation lubricated with poly(ADP-ribose)

Watanabe and colleagues have largely addressed the concerns that I raised in the initial review, and they have added additional experiments to address the identity of the PARP responsible for producing poly(ADP-ribose). Although the responsible PARP still remains to be identified, I feel that they have added the appropriate comments to the text, and they have added to the Supplement the work that they have performed thus far.

Response:

We appreciate the reviewer's consideration for the limitation in this study.

Regarding point-by-point Fig. 2 that shows that RNA can inhibit ASK3 condensates in vitro, I feel that this data could be included in the study, but perhaps with a control experiment that shows that some length of RNA (or even just ribonucleotides) does not have the effect. And perhaps whether uncharged polymers, or positively charged polymers, have the same effect.

Response:

We appreciate the reviewer's advice. We re-performed this experimental set with a negative control sample for RNA addition (i.e., addition of adenosine 5'-monophosphate (AMP)) and incorporated the result in our revised manuscript (lines 261–264 of page 7, Supplementary Fig. 7). As the reviewer suggested, the uncharged and positively charged polymers are ideal for the comparisons; however, we consider that the appropriate polymers corresponding to PAR/RNA are not easily defined. For instance, polyethylene glycol (PEG) does not resemble PAR/RNA but is an uncharged polymer at least, and PEG is added as an inducer in our assay. Hence, the affinity of prepared polymers with ASK3 would affect the interpretation of results, and we did not prepare these ideal control samples. In this revised manuscript, we did not specifically describe the commonality between PAR and RNA as “negatively charged property” nor “electrostatic interactions”; that is, we fairly described just “the common physicochemical property between PAR and RNA” (line 263 of page 7), although the experiments based on PAR-binding motif (PBM) also implies the importance of negatively charged property of PAR.

Reviewer #2 (Remarks to the Author):

The authors have fully and thoughtfully addressed my comments in the revised manuscript.

Stefan Hohmann

Response:

We thank you for your time to review our revised manuscript.

Reviewer #3 (Remarks to the Author):

Watanabe et al. carefully addressed the review comments during revision, providing new data in both the manuscript and the point-by-point reply. In my opinion, there are three key questions that came up during the revision:

1. Is the work still timely in light of recent publications?
2. Did the authors uncover the full mechanism?
3. Does the LLPS model of the authors for ASK3 regulation under hypertonic stress add up?

My take is:

1. Despite multiple recent reports of ‘hyperosmotic phase separation (HOPS)’, I think that the work of Watanabe et al. stands out because of their efforts to uncover the mechanism and functional meaning of (ASK3) HOPS, rather than merely describing the phenomenon. Thus, I still consider the work of Watanabe et al. timely, novel and exciting (perhaps it is also noteworthy in this regard that they shared their findings as a preprint approx. one month earlier than Jalihal et al.).

Response:

We are relieved to hear the reviewer’s fair evaluation. Thank you.

2. There has been great excitement about the manuscript during review, but also a consensus that the mechanisms has not yet been fully worked out. An obvious missing piece to the puzzle is the identity of the PARP that modifies ASK3 condensates. I do not think that the bar for publication should be that the full regulatory pathway of a complex biological event, including all molecular players, has to be worked out in a single paper. The authors provide plenty of exciting and stimulating findings of interest to many fields (stress response biology, condensate function, kinase regulation, PARYlation, ...) that will move science—a collective endeavor—forward.

Response:

We appreciate and agree with the reviewer’s thought.

3. This leaves the most important question: is the current working model of the authors plausible, i.e. backed by the data? To me, a key aspect of their model that hadn’t been directly tested in the original manuscript is the whether PAR-dependent tuning of the ASK condensate material properties is truly required for PP6 recruitment to the condensate surface. In response to my comment, the authors performed an experiment with a perhaps unexpected, but nevertheless exciting result: PP6 is fully dispersed in PBM4 mutant condensates. This finding is very intriguing, but brings up a number of questions:

3a. Given that PBM4 mutant droplets are more solid, this suggests that the whole regulatory system may now be trapped in a dead-end aggregate (this could be further tested by two-channel FRAP). Such aggregates may work fine to inactive ASK3 in response to a singular hypertonic insult even without complete dephosphorylation, but I wonder if the cells now lose their ability to respond to hypotonic stress – such as the cells experience when returning to isotonic from hypertonic?

Response:

We appreciate the reviewer’s comment. To exclude the possibility that the ASK3 PBM4 mutant forms

mere artifact aggregates under hyperosmotic stress, we investigated the reversibility of ASK3 PBM4 condensates. When setting back to isoosmotic conditions, the hyperosmotic stress-induced condensates of the PBM4 mutant disappeared (2nd point-by-point response Figure). Hence, the possibility that the whole regulatory system is just trapped and dephosphorylation of the ASK3 PBM4 mutant is prevented would be unlikely.

2nd point-by-point response Figure. Reversibility of ASK3 PBM4 condensates in EGFP-FLAG-ASK3(PBM4)-transfected HEK293A cells. After hyperosmotic stress (600 mOsm, 20 min), the extracellular osmolality was set back to the isoosmotic condition. White bar: 20 μ m.

3b. I also wonder what FK866 treatment does to wild type ASK3-PP6 condensates? I'm not sure that I agree that FK866 treatment truly "inhibited [...] the interaction between PP6 and ASK3 under hypertonic stress" (lines 226-228) based on the data shown in Fig.4g – it rather looks slightly reduced at best. Perhaps the phenotype on the condensate level is the same as for PBM4 and what is really going on is that lack of PAR causes ASK3 and PP6 to 'aggregate'?

Response:

We appreciate the reviewer's thoughtful comment. We agree that the effect of FK866 pretreatment on PP6-ASK3 interaction is relatively slight in Fig. 4g, but we could conclude this slight effect with confidence because of the inverse effect of NMN pretreatment (Fig. 4h). In contrast, we have not been able to confirm the interaction between PP6 and the PBM4 mutant from another viewpoint yet, and we did not conclude the result as described in the previous point-by-point response. Nevertheless, as the reviewer suggested, we consider that FK866-pretreated ASK3 condensates have basically same characteristics with ones of ASK3 PBM4 condensates. In fact, we performed a microscopic experiment of FK866 pretreatment and found that FK866 pretreatment makes ASK3 condensates merge with PP6 condensates but instead lose the shared phase boundary (Supplementary Fig. 8b), which is the similar pattern with condensates of the PBM4 mutant (Supplementary Fig. 8a). We added this result to our revised manuscript (lines 392-395 of page 9, Supplementary Fig. 8b).

3c. Thus, I still think that directly showing, on the condensate level by microscopy, that PARylation alters ASK3 and PP6 interaction would be required to make the condensate-based model of the authors fully convincing – especially as the authors themselves cast doubt on the IP approach when they speculate about "the difficulty of holding the PP6-ASK3 interaction during in vitro operations", which is a valid concern with condensate-based interactions.

Response:

We believe that the reviewer's concerns are relieved by the above responses to point 3a and point 3b.

REVIEWERS' COMMENTS

Reviewer #1 (Remarks to the Author):

In the revised manuscript, the authors have satisfied the additional point that I raised with a new experiment and associated text.

Reviewer #3 (Remarks to the Author):

Watanabe et al. have further refined their beautiful work, although I'm not quite sure why they bury their intriguing new multi-phase condensate data in the supplements, especially given its centrality to their model.

However, I'm still adamant that the statement in lines 386-388 has to be revised prior to publication. PAR depletion did not inhibit the interaction between PP6 and ASK3 under hypertonic stress, even if NMN pre-treatment enhanced it. This conclusion is neither backed by the data, nor does it fit into the model of Watanabe et al. that the key regulatory step occurs at a phase boundary. PP6 and ASK3 clearly still co-localize in the condensates (Supp. Fig. 8), and the IP in Fig. 4g suggests that they also still interact physically within there. Besides, as pointed-out repeatedly during the revisions, classical IPs are not well-suited to probe droplet interactions – thus, the authors should not draw such bold conclusions from their IP data. At the very least, the verb “inhibited” in line ought to be changed to “reduced”.

Point-by-point response

REVIEWER COMMENTS

Reviewer #1 (Remarks to the Author):

In the revised manuscript, the authors have satisfied the additional point that I raised with a new experiment and associated text.

Response:

We appreciate your time to review our revised manuscript.

Reviewer #3 (Remarks to the Author):

Watanabe et al. have further refined their beautiful work, although I'm not quite sure why they bury their intriguing new multi-phase condensate data in the supplements, especially given its centrality to their model.

However, I'm still adamant that the statement in lines 386-388 has to be revised prior to publication. PAR depletion did not inhibit the interaction between PP6 and ASK3 under hypertonic stress, even if NMN pretreatment enhanced it. This conclusion is neither backed by the data, nor does it fit into the model of Watanabe et al. that the key regulatory step occurs at a phase boundary. PP6 and ASK3 clearly still co-localize in the condensates (Supp. Fig. 8), and the IP in Fig. 4g suggests that they also still interact physically within there. Besides, as pointed-out repeatedly during the revisions, classical IPs are not well-suited to probe droplet interactions – thus, the authors should not draw such bold conclusions from their IP data. At the very least, the verb “inhibited” in line ought to be changed to “reduced”.

Response:

We appreciate the reviewer's thoughtful comments. In contrast to the reviewer's interpretation for Fig. 4g, we basically interpret that the FK866 pretreatment “slightly reduces” the PP6–ASK3 co-immunoprecipitation under hyperosmotic stress, which is based on our multiple results including different experimental conditions. As the reviewer pointed out, this interpretation may seem to be contradictory; that is, the physical interaction between PP6 and ASK3 is “reduced” (Fig. 4g), while ASK3 almost completely colocalizes with PP6 under FK866 pretreatment (Supplementary Fig. 8b). However, the colocalization observed in conventional fluorescence microscopy does not necessarily indicate the physical molecule–molecule interaction because we can only detect the expanded fluorescent signals as a single sub-micrometer dot. Therefore, as one of coherent explanations for the seemingly contradictory findings, we hypothesize that the functional physical interactions between PP6 and ASK3 occur only in the surface vicinity of condensates and that the PP6–ASK3 interactions within the core of condensates do not occur or occur just as pseudo stochastic interaction. To prove this hypothesis, we would have to perform single-molecular analysis; namely, we speculate that the PP6–ASK3 colocalization is observed only around the sharing surface vicinity of condensates at the single-molecular level resolution, although we believe that the single-molecular imaging technique is a task beyond this study. Given this current interpretation, we did not include Supplementary Fig. 8 as the main figure. Nevertheless, we understand the reviewer's interpretation; and we completely agree that the verb “inhibited” was too strong. In this revised manuscript, we toned down the potential reduction of PP6–ASK3 interaction by FK866 pretreatment (yellow-highlighted line 223 in page 6 and line 383 in page 9).

Additionally, as we mentioned in the 1st point-by-point response to the reviewer #3, we take the same

position with the reviewer; i.e., classical immunoprecipitation and pull-down assays are not necessarily suitable for study on liquid-phase condensates. However, they have been leveraged to suggest the interactions within the condensates in many papers (e.g., Rai, A. K. et al. *Nature* 2018). Hence, we can easily imagine that a graduate student gets in trouble by classical PI in biochemistry when he or she obtains the true but seemingly contradictory results between pull-down-based and microscopy-based experiments. Because our point-by-point response is to be alongside main article online in *Nature Communications*, Fig. 4g would be also worth being presented in point of helping the diligent students.